# ACTIVE-O3: EMPOWERING MLLMS WITH ACTIVE PERCEPTION VIA PURE REINFORCEMENT LEARNING

## ABSTRACT

Active vision, also known as active perception, refers to actively selecting where and how to look in order to gather task-relevant information. It is a critical component of efficient perception and decision-making in humans and advanced embodied agents. With the rise of Multimodal Large Language Models (MLLMs) as central planners in robotic systems, the lack of methods for equipping MLLMs with active perception has become a key gap. We first provide a systematic definition of MLLM-based active perception tasks and show that GPT-o3's zoom-in strategy can be viewed as a special case, though it suffers from low efficiency and inaccurate region selection. To address these issues, we propose **ACTIVE-O3**, a reinforcement learning framework built on GRPO that equips MLLMs with active perception capabilities. Leveraging a modular sensing–action design and a dual-form reward, **ACTIVE-O3** autonomously learns efficient and stable region selection strategies without explicit supervision. We further establish a comprehensive benchmark covering both open-world tasks (small/dense-object grounding) and domain-specific scenarios (remote sensing, autonomous driving, interactive segmentation). Experimental results demonstrate that ACTIVE-O3 significantly enhances active perception capabilities compared to Qwen2.5-VL-CoT. Moreover, we show that our RL framework not only preserves the model's general understanding ability but can also serve as a proxy task for leveraging perception data, further improving performance on benchmarks such as RealWorldQA and MME. We hope that our work can provide a simple codebase and unified evaluation protocol to facilitate future research on *active perception MLLM*.

## 1 INTRODUCTION

*"We must perceive in order to move, but we must also move in order to perceive."*
                                        — James J. Gibson, *The Ecological Approach to Visual Perception* (1979)

Among the many components of perception, active perception, the process of selective acquisition of sensory information to achieve specific goals, has proven essential for efficient information gathering and decision making in complex environments (Aloimonos et al., 1988; Ballard, 1991; Whaite & Ferrie, 1997). For humans, active perception enables tasks such as focusing on relevant details in a cluttered scene or dynamically adjusting viewpoints to better understand ambiguous objects. Similarly, embodied agents, such as autonomous robots, must also make intelligent choices about where to look and how to look to succeed in real-world tasks (Arruda et al., 2016; Das et al., 2018; Chaplot et al., 2020).

With the recent surge in the capabilities of multimodal large language models (MLLMs) (Achiam et al., 2023; Jaech et al., 2024; Liu et al., 2024a; Bai et al., 2025), these models are increasingly being integrated into robotic systems (Qi et al., 2025b; Yang et al., 2025; Team et al., 2025; Kim et al., 2024; Black et al., 2024; Intelligence et al., 2025) as central modules for planning, reasoning, and decision-making. However, despite their impressive generalization and compositionality, current MLLMs are typically passive consumers of visual inputs, relying on static, fixed views of the environment. This contrasts sharply with the dynamic information-seeking behavior that characterizes active perception. A recent attempt to move towards active perception in MLLMs is the *zoom-in search strategy proposed in GPT-o3*. Although this strategy offers a first step, it remains limited by inefficient region proposals and low target localization accuracy(see Figure 4 in Appendix), espe-

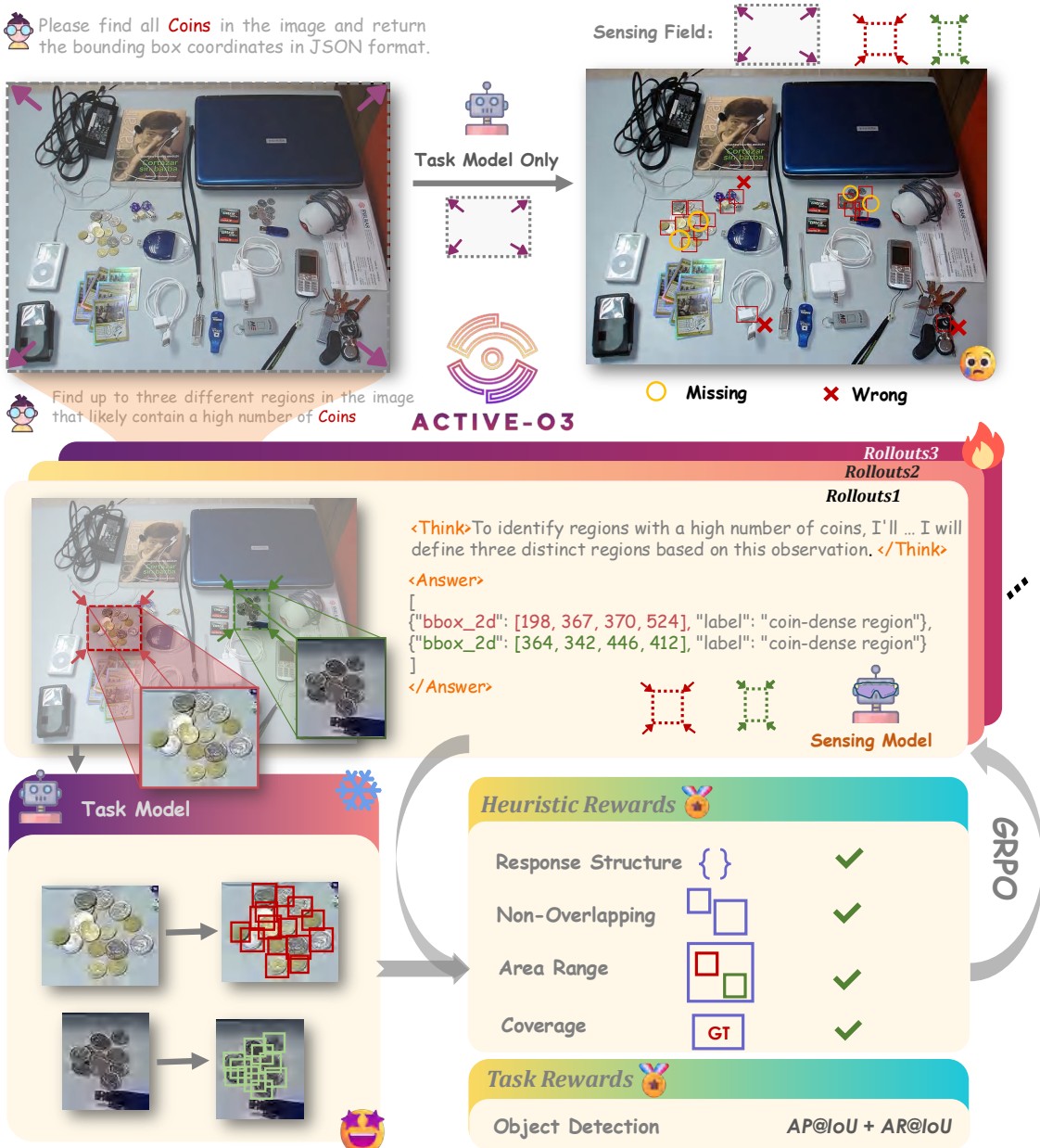

Figure 1: Overview of the proposed Active-O3 framework. Given a multimodal query (e.g., "find all coins"), traditional task models often miss or misidentify target objects due to limited perceptual coverage. Active-O3 enhances perception by allowing the model to actively propose informative subregions (zoom-in regions) based on a learnable sensing policy. For clarity of visualization, we only display two zoom-in regions in this figure, although the model can propose up to three regions.

cially in dense or fine-grained scenarios. Crucially, there is still a lack of systematic frameworks and evaluation protocols to study and develop active perception capabilities within MLLMs.

In this paper, we present **ACTIVE-O3**, a reinforcement learning–based training framework built on Group Relative Policy Optimization (GRPO) (Guo et al., 2025), specifically designed to endow MLLMs with active perception capabilities. We first provide a formal task definition of MLLM-based active perception, and implement it by decoupling a single MLLM backbone into a sensing module and a task (action) module. Unlike prior search-based approaches, our model autonomously learns efficient parallel region selection strategies during RL training.

In this paper, we present **ACTIVE-O3**, a reinforcement learning–based training framework built on Group Relative Policy Optimization (GRPO) (Guo et al., 2025), focusing on the 2D active perception setting for MLLMs, where reproducibility and controlled evaluation are feasible. We first provide a formal task definition of MLLM-based active perception, which aims to offer a general and modular foundation rather than to cover all forms of embodied active perception, and implement it by decoupling a single MLLM backbone into a sensing module and a task (action) module. Unlike prior search-based approaches, our model autonomously learns efficient parallel region selection strategies during RL training, a design choice made for efficiency and coverage under fixed sensing budgets. Importantly, although we adopt GRPO as the default policy optimization algorithm, our experiments show that the framework is algorithm-agnostic and can achieve similar gains with multiple reinforcement learning algorithms, demonstrating that the key contribution lies in the proposed active perception framework rather than in any specific algorithm.

We further observe that relying solely on a task-oriented reward is too sparse and easily dominated by the action module, which hinders the learning of diverse and reasonable sensing strategies. To address this, we design a *dual-form reward* that combines a task-aware component with a heuristic component, thereby enhancing both stability and effectiveness in RL training. To comprehensively evaluate performance, we construct a benchmark suite covering a broad spectrum of tasks—ranging from open-world grounding of small and dense objects, to domain-specific applications such as remote sensing, autonomous driving, and fine-grained segmentation. Extensive experiments demonstrate that ACTIVE-O3 consistently improves search efficiency, accuracy, and downstream task performance under fixed computational budgets. Moreover, we find that, despite not being explicitly trained on reasoning or QA data, ACTIVE-O3 preserves general understanding and reasoning abilities, and even surpasses our initialization model Qwen2.5-VL-7B on RealWorldQA, MME, and MMVU. This shows that active perception can serve as a general proxy objective, effectively leveraging perception annotations to enhance the visual understanding and reasoning capabilities of MLLMs.

Our primary contributions are summarized as follows:

- We propose ACTIVE-O3, the first reinforcement learning framework for active perception with MLLMs, formalized via a unified modularly decoupled policy that separates region proposal (sensing) and task execution. Our method combines structured instruction prompts with a dual-form reward design—integrating both task-aware and heuristic feedback—to guide the model toward producing diverse, interpretable, and task-effective region proposals.

- We target two representative yet challenging applications—namely, small/dense object detection and interactive segmentation—and demonstrate that ACTIVE-O3 significantly improves perception quality and task performance across both general-purpose and domain-specific visual tasks.

- We show that despite not being trained on reasoning or QA data, ACTIVE-O3 preserves and even enhances general understanding and reasoning abilities, outperforming its initialization model on RealWorldQA, MME. This demonstrates that active perception can serve as a general proxy task, leveraging perception annotations to improve MLLMs' visual understanding and reasoning.

## 2 RELATED WORKS

### 2.1 REINFORCEMENT LEARNING FOR MULTIMODAL LARGE LANGUAGE MODELS

While supervised learning and instruction tuning remain the dominant approaches for training MLLMs, several limitations persist—such as aligning model behavior with human preferences and handling complex reasoning tasks. Reinforcement Learning (RL) has been introduced as a promising approach to address these challenges. An early and influential example is Reinforcement Learning from Human Feedback (RLHF) (Ouyang et al., 2022), which was primarily developed to align model behavior with human preferences and played a central role in the success of ChatGPT (Achiam et al., 2023). A recent advancement in this direction is Group Relative Policy Optimization (GRPO), proposed in DeepSeek-R1 (Guo et al., 2025) and DeepSeek-Math (Shao et al., 2024). GRPO introduces a novel way to estimate the advantage function using the mean and variance of rewards across a group of responses, guided by verifiable reward signals. This approach eliminates the need for a separate critic model and significantly enhances reasoning capabilities on complex problems. Concurrently, several works (Zhao et al., 2025; Feng et al., 2025; Liu et al.,

2025; Huang et al., 2025; Shen et al., 2025a) have explored applying GRPO to MLLMs. However, these efforts mainly focus on text-centric reasoning or simple visual grounding tasks. In contrast, our work investigates how GRPO can empower MLLMs with active perception abilities, targeting visually grounded reasoning tasks that require spatial understanding and goal-directed attention. Moreover, due to the difficulty of collecting high-quality trajectories for active perception scenarios, reinforcement learning becomes even more essential in this context.

## 2.2 Active Perception

Active perception refers to the paradigm in which an agent intelligently and dynamically controls its own sensors or actions to achieve a specific task or goal. Early foundational work (Aloimonos et al., 1988; Ballard, 1991; Whaite & Ferrie, 1997)—often termed "active vision" when focusing on visual sensors—demonstrated that by actively controlling parameters such as camera pose or sensor configuration, agents can transform otherwise ill-posed perception problems into well-posed ones. This enables more efficient and effective information gathering for tasks like object recognition, scene understanding, navigation, and manipulation. More recently, the principles of active perception have been widely embraced in the field of embodied AI (Arruda et al., 2016; Das et al., 2018; Chaplot et al., 2020; Jayaraman & Grauman, 2018), where agents must not only perceive but also interact purposefully with their environments to accomplish complex goals. Meanwhile, there is a clear trend toward integrating Multimodal Large Language Models (MLLMs) as the central reasoning modules—or "brains"—of embodied AI systems (Black et al., 2024; Kim et al., 2024; Qi et al., 2025a). In this context, enabling MLLMs with active perception capabilities is of critical importance for advancing the autonomy and intelligence of such systems. However, despite rapid progress in MLLM research, active perception remains largely underexplored. Our work aims to bridge this gap, leveraging the strong generalization and reasoning capabilities of MLLMs to tackle challenges in active perception.

## 2.3 Visual CoT

Recent works such as Visual CoT prompting (Chen et al., 2024), ARGUS (Man et al., 2025), Re-Focus (Fu et al., 2025), Chain-of-Spot (Liu et al., 2024d), and GRIT (Fan et al., 2025) improve multimodal reasoning by generating grounded reasoning chains or editing visual evidence, yet they operate on a fixed image and remain reasoning-centric. ZoomEye (Shen et al., 2025b) explores zooming through heuristic tree search, while DeepEyes (Zheng et al., 2025) uses RL to optimize visual thinking, but neither learns a general sensing policy. These approaches assume the relevant region is already visible and focus on producing better answers. Compared with recent Visual CoT and visual grounding works, our approach differs in three fundamental ways: (1) **Region exploration**: Visual CoT typically performs localized object grounding, while our setting requires exploratory region search under uncertainty, especially when objects are hard to detect or not initially visible. (2) **Task focus**: Prior works target reasoning-centric tasks (e.g., QA, attribute reasoning) that rely on existing grounding ability, whereas we study perception-centric tasks such as small-object detection and dense-scene understanding that demand genuine active sensing. (3) **Methodology**: Visual CoT methods rely on prompting or supervised fine-tuning with annotated reasoning chains or boxes, while our framework learns a sensing policy via pure reinforcement learning without extra supervision, with explicit modular decoupling between sensing and task modules.

## 3 MLLM-based Active Perception: Definition and Analysis

In this section, we provide a formal definition of active perception tasks based on multi-modal large language models (MLLMs) (see Figure 1 for our framework and more analysis in Appendix D).

**Modular View of Active Perception.** Consider an embodied agent that receives a human instruction $\mathcal{I}$ and is required to perform a complex physical-world task. At each time step $t$, the agent state is defined as $s_t = (s_t^{\text{env}}, s_t^{\text{cam}})$, where $s_t^{\text{env}}$ describes the environment (e.g., objects and their properties), and $s_t^{\text{cam}}$ denotes the sensor's pose and viewpoint. A deterministic observation function $g$ maps the current system state to a visual observation: $o_t = g(s_t) + \epsilon_t$ where $\epsilon_t$ is a stochastic noise term. The action space is similarly factorized as $a_t = (a_t^{\text{env}}, a_t^{\text{cam}}) \in \mathcal{A}$, where $a_t^{\text{env}}$ denotes

**Prompt for** ACTIVE-O3 **Detection**

- "Find up to three different regions in the image that likely contain a high number of '**{object}**'."
- "Even if the '**{object}**' are not clearly visible, infer where they are most likely to appear."
- "Each region should cover multiple '**{object}**' and include some visual context."
- "The selected regions should be as distinct as possible, with minimal or no overlap between them."
- "Return the coordinates in JSON format as: {"bbox_2d": [x1, y1, x2, y2], "label": "**{object}**-dense region"}."
- "Explain your reasoning in `<think>`...`</think>` and output the final result in `<answer>`...`</answer>`."
- "Example: `<think>` thinking process here `</think>` `<answer>` JSON format here `</answer>`"

Figure 2: Prompt for ACTIVE-O3-DET.

the task-oriented interaction action (e.g., grasping, pointing), and $a_t^{\text{cam}}$ controls the sensing parameters (e.g., moving or rotating the camera). In order to effectively interact with the environment, the agent must continuously adjust its visual perspective based on current observations to acquire more informative inputs that guide subsequent actions. Active perception can thus be modeled as a coordination between two modules:

- **Task Model** $\mathcal{M}_A$: decides how to act on the environment to accomplish external tasks. It takes the current observation $o_t$ and the task instruction $\mathcal{I}$ as input, and outputs a task-level action:

$$a_t^{\text{env}} = \mathcal{M}_A(o_t, \mathcal{I})$$

- **Sensing Model** $\mathcal{M}_O$: decides how to control perception parameters to improve observation quality. It also takes the current observation and task instruction as input, and outputs a perception action:

$$a_t^{\text{cam}} = \mathcal{M}_O(o_t, \mathcal{I})$$

In our formulation, each action component primarily affects a specific part of the system state: $a_t^{\text{cam}}$ updates $s_t^{\text{cam}}$, and $a_t^{\text{env}}$ updates $s_t^{\text{env}}$, formalized as

$$s_{t+1}^{\text{cam}} = f^{\text{cam}}(s_t^{\text{cam}}, a_t^{\text{cam}}), \quad s_{t+1}^{\text{env}} = f^{\text{env}}(s_t^{\text{env}}, a_t^{\text{env}})$$

where $f^{\text{cam}}$ and $f^{\text{env}}$ are deterministic transition functions. At each time step, the system operates in a closed loop as follows: 1) the sensing model selects a perception action $a_t^{\text{cam}} = \mathcal{M}_O(o_t^{\text{prev}}, \mathcal{I})$, which updates the sensor state via $s_t^{\text{cam}} \leftarrow f^{\text{cam}}(s_t^{\text{cam}}, a_t^{\text{cam}})$; 2) the system receives a new observation $o_t = g(s_t) + \epsilon_t$; 3) based on $o_t$ and $\mathcal{I}$, the action model selects an interaction action $a_t^{\text{env}} = \mathcal{M}_A(o_t, \mathcal{I})$, which updates the environment state as $s_{t+1}^{\text{env}} = f^{\text{env}}(s_t^{\text{env}}, a_t^{\text{env}})$.

**Objective Function** We jointly optimize the action model $\mathcal{M}_A$ and the sensing model $\mathcal{M}_O$ to maximize task success while minimizing perceptual cost:

$$\max_{\mathcal{M}_A, \mathcal{M}_O} \mathbb{E}\left[\sum_{t=1}^{T} R(s_t, a_t^{\text{env}}) - \lambda \cdot C(a_t^{\text{cam}})\right]$$

where $R(s_t, a_t^{\text{env}})$ denotes the task-level reward (e.g., success or progress), $C(a_t^{\text{cam}})$ is the cost of the sensing action (e.g., viewpoint shift or latency), and $\lambda$ is a balancing factor.

## 4 ACTIVE-O3

We present ACTIVE-O3, a unified framework for MLLM-driven active perception in vision–language tasks. Our goal is to provide a principled and modular formulation of active perception for MLLMs, serving as a conceptual foundation for future research, rather than aiming to

cover all forms of embodied active perception. Although our formulation is general and applies to embodied agents in complex physical environments, such real-world 3D settings hinder reproducible deployment and fair evaluation. We therefore instantiate the framework in a controlled, fair, and reproducible 2D setting: active perception over static images. Within this setting, we study two challenging perception-centric applications—(i) small-object detection/grounding and (ii) interactive segmentation—both of which require selecting multiple informative regions before executing task-specific actions. This instantiation follows directly from the general formulation while allowing us to isolate the sensing policy and rigorously evaluate MLLM-based active perception under fixed sensing budgets.

Given an image $I$ and instruction $\mathcal{I}$, we first generate a global observation $o_{\text{init}}$ by resizing $I$. A shared multi-modal large language model (MLLM) is treated as a unified policy $\pi$ that generates a textual response $y$—containing both intermediate reasoning and action outputs—conditioned on the visual input and instruction, i.e., $\pi(y \mid o, \mathcal{I})$. The MLLM is then guided by two prompts: $\mathcal{I}_O$ for proposing regions, and $\mathcal{I}_A$ for performing task-specific operations. We extract actionable components from $y$ via task-specific parsers tailored to each subtask. In this setup:

- **Sensing module:**
$$\mathcal{M}_O(o_{\text{init}}, \mathcal{I}_O) := \text{Parse}_{\text{cam}}(\pi(y \mid o_{\text{init}}, \mathcal{I}_O))$$
which produces $K$ candidate perception actions $\{a_k^{\text{cam}}\}_{k=1}^K$ parsed from the full response.

- **Task module:**
$$\mathcal{M}_A(o_k, \mathcal{I}_A) := \text{Parse}_{\text{env}}(\pi(y \mid o_k, \mathcal{I}_A))$$
which operates on the $k$-th region crop and produces the final task-level output $a_k^{\text{env}}$.

## 4.1 Objective in 2D Active Perception with Parallel Region Selection

A key property of the 2D visual scenario is that the environment state $s^{\text{env}}$ remains static across time (since the interaction action $a_t^{\text{env}}$ does not change the image). In this setting, we fix the task model $\mathcal{M}_A$ and focus on learning a sensing policy $\mathcal{M}_O$ that selects a set of $K$ informative regions from a static image $I$, conditioned on an initial observation $o_{\text{init}}$ and instruction $\mathcal{I}$. Here, $o_{\text{init}}$ is a low-resolution global view (e.g., a thumbnail), which serves as a coarse prior for guiding more detailed inspection. The goal is to maximize task performance under a fixed sensing budget. Formally, the optimization objective is:

$$\max_{\mathcal{M}_O} \mathbb{E}_{I,\mathcal{I}} \left[ \sum_{k=1}^K R\left(\mathcal{M}_A\left(o_k\right), \mathcal{I}\right) \right], \quad \text{where} \begin{cases} \{a_k^{\text{cam}}\}_{k=1}^K = \mathcal{M}_O(o_{\text{init}}, \mathcal{I}) \\ o_k = \text{ResizeCrop}(I, a_k^{\text{cam}}) \end{cases} \tag{1}$$

We formulate active perception in this static 2D case as a single-step decision problem ($T = 1$). Crucially, instead of sequentially zooming in region by region, the policy produces *parallel region selections* $\{a_k^{\text{cam}}\}_{k=1}^K$ in one forward pass. This design encourages the policy to generate a diverse yet complementary set of regions, improving coverage and efficiency under the sensing budget. In detection-style tasks, $a_k^{\text{env}}$ shares the same format as $a_k^{\text{cam}}$—a bounding box list—but differs in role: $a^{cam}$ proposes regions for inspection, while $a^{env}$ expresses final localization predictions. We evaluate alignment between $a_{1:K}^{\text{env}}$ and ground-truth boxes $\text{GT}_{\text{box}} = \{(x_1, y_1, x_2, y_2)\}$ using standard metrics such as Average Precision (AP) and Average Recall (AR).

## 4.2 Initial Sensing Policy via MLLM

To enable active perception without additional supervised fine-tuning (SFT), we leverage the instruction-following and reasoning abilities of MLLMs to construct an *initial sensing policy $\mathcal{M}_O$* via prompting. This zero-shot initialization provides two advantages: (i) it offers a practical way to bootstrap active perception without task-specific labels, and (ii) it naturally aligns with our modular decoupling design, where sensing (region proposal) and acting (task execution) are separated. Importantly, the prompt-based policy generates both interpretable reasoning traces and candidate regions $a_{1:K}^{\text{cam}}$, serving as a transparent and effective starting point for subsequent RL optimization.

We design a task-specific instruction prompt $\mathcal{I}_O$ (Figure 2) to guide $\mathcal{M}_O$ in producing meaningful and diverse region proposals $a_{1:K}^{\text{cam}}$. The prompt serves three key purposes:

- **Format regularization:** The prompt enforces a structured output format and encourages step-by-step reasoning using tags such as `<think>` and `<answer>`.
- **Task guidance:** It introduces domain-specific priors, such as encouraging the model to:
  - infer likely object locations even when objects are not clearly visible,
  - select spatially diverse and minimally overlapping regions,
  - prefer regions with sufficient surrounding context to support downstream decisions.

These constraints help $\mathcal{M}_O$ generate interpretable and effective sensing actions that form the basis for active region selection.

## 4.3 Policy Improvement with GRPO

While the prompt-based policy provides a reasonable starting point, it cannot adapt to task-specific feedback: the value of a region proposal $a^{\mathrm{cam}}$ depends entirely on its downstream effect via $\mathcal{M}_A$, which makes supervised targets unavailable. Moreover, we desire $\pi(y \mid o_{\mathrm{init}}, \mathcal{I}_O)$ to output not only $K$ parallel region proposals but also coherent reasoning traces, further complicating direct supervision. To address this, we improve $\mathcal{M}_O$ using reinforcement learning with task-level rewards. Specifically, we adopt **GRPO**, a lightweight policy optimization method that avoids training a critic model and is thus well-suited for large language models. This enables the sensing policy to refine its parallel selection strategies end-to-end, guided solely by the effectiveness of the downstream task performance.

Let $\pi_\theta$ denote the current policy and $\pi_{\theta_{\mathrm{old}}}$ the behavior policy used to sample $N$ responses $\{y_n\}_{n=1}^N$. Each response contains reasoning and candidate region proposals parsed as $a_{1:K}^{\mathrm{cam}} = \mathrm{Parse}_{\mathrm{cam}}(y_n)$. The training objective is:

$$\mathcal{J}_{\mathrm{GRPO}}(\theta) = \mathbb{E}_{I,\mathcal{I}} \left[ \frac{1}{N} \sum_{n=1}^N \min\left(w_n(\theta)\, A_n,\ \mathrm{clip}(w_n(\theta),\ 1-\epsilon,\ 1+\epsilon)\, A_n\right) - \beta\, \mathbb{D}_{\mathrm{KL}}(\pi_\theta \| \pi_{\mathrm{ref}}) \right] \tag{2}$$

where $w_n(\theta) = \frac{\pi_\theta(y_n | o_{\mathrm{init}}, \mathcal{I}_O)}{\pi_{\theta_{\mathrm{old}}}(y_n | o_{\mathrm{init}}, \mathcal{I}_O)}$ is the importance ratio between current and behavior policies, $A_n$ is a normalized reward-based advantage for sample $n$, and $\pi_{\mathrm{ref}}$ is a frozen reference policy (e.g., the base MLLM) used to regularize the update.

$$A_n = \frac{r_n - \mathrm{mean}(\{r_1, \ldots, r_N\})}{\mathrm{std}(\{r_1, \ldots, r_N\})} \tag{3}$$

## 4.4 Dual-Form Reward Design

A central novelty of our framework lies in the *dual-form reward*, which jointly leverages task-independent heuristics and task-specific feedback. This design addresses a key challenge in active perception: purely task-based rewards are often too sparse and unstable, while purely heuristic rewards may fail to align with downstream objectives. By combining both, our method provides dense, interpretable signals to stabilize training, while still driving the policy toward end-task performance(see detailed ablation results in Table 12 of Appendix).

**Heuristic Reward.** This reward evaluates a single MLLM response based on task-independent criteria that promote interpretable and spatially meaningful region proposals. It is composed of four components:

- **Format Validity.** The response must conform to a valid structured format. We reward responses that are parseable as JSON with bounding boxes under the `bbox_2d` field and that include both reasoning and answer segments marked by `<think>` and `<answer>` tags.
- **Non-overlapping Proposals.** To encourage spatial diversity, we reward proposals whose pairwise Intersection-over-Union (IoU) falls below a threshold. Responses with any overlapping regions are penalized.
- **Area Range Constraint.** Each bounding box is required to fall within a reasonable size range relative to the image (e.g., 1% to 50%). This avoids overly small or overly large boxes that may be either noisy or uninformative.

- **Coverage-Based Reward.** When ground truth masks or boxes are available, we assess how well the predicted regions align with task-relevant areas. This can include: (i) the proportion of ground-truth mask pixels covered by a region, (ii) the percentage of ground-truth boxes matched by at least one proposal, or (iii) the Dice/IoU between predicted and reference masks.

The final heuristic reward $\mathcal{R}_{\text{heuristic}}(y)$ is computed as a weighted sum of the above components.

**Task-Aware Reward.** The task-aware reward evaluates the quality of the selected regions based on their downstream utility as measured by task-specific performance metrics. To compute this reward, we execute the task model $\mathcal{M}_A$ on each selected region $o_k$, generating outputs $a_k^{\text{env}}$. This requires additional forward passes of $\mathcal{M}_A$ during training, for which we implement an efficient batched inference system to support parallel evaluation.

The form of the reward depends on the specific task:

- **Detection:** $\mathcal{M}_A$ returns a set of predicted bounding boxes $\{b_i\}_{i=1}^K$, which are compared against ground-truth boxes $\{b_j\}_{j=1}^J$ using standard metrics such as Average Precision (AP) and Average Recall (AR), based on IoU matching.
- **Interactive Segmentation:** $\mathcal{M}_A$ predicts interaction points (positive/negative) based on each region, which are fed to a local instance of Segment Anything (SAM) via an internal API. The resulting segmentation mask is compared against ground-truth masks using mean Intersection over Union (mIoU).

Together, the dual-form reward balances **stability and task alignment**, enabling ACTIVE-O3 to learn active perception strategies that are both interpretable and effective. Formal definitions and implementation details are provided in Appendix Sections B and C.

## 5 EXPERIMENTS

### 5.1 COMPARED METHODS

In this section, we introduce three baseline methods and a variant of ACTIVE-O3 to conduct a comparison (see Figure 10 for visualization results, and more ablation results can be found in the Appendix).

- **Grounding DINO (GDINO)** (Liu et al., 2024b): A strong open-world detection model used as $\mathcal{M}_A$, which performs grounding directly without $\mathcal{M}_O$.
- **Qwen2.5-VL-7B** (Bai et al., 2025): Adopted as an MLLM-based task model $\mathcal{M}_A$, evaluating pure MLLM performance on grounding without auxiliary $\mathcal{M}_O$.
- **Qwen2.5-VL-CoT**: As in Sec. 4.2, we reuse Qwen2.5-VL-7B both as $\mathcal{M}_O$ (action proposals) and $\mathcal{M}_A$ (action execution).
- **ACTIVE-O3 + GDINO**: We decouple $\mathcal{M}_A$ and $\mathcal{M}_O$ at test time, replacing $\mathcal{M}_A$ with GDINO while retaining $\mathcal{M}_O$ from ACTIVE-O3. This configuration tests whether ACTIVE-O3's sensing policy can generalize when paired with a stronger, specialized task model.
- **V\* + GDINO**: V\* (Wu & Xie, 2024) is a typical MLLM-based search algorithm. We use its generated search trajectory as the output of the sensing module $\mathcal{M}_O$. Notably, such search-based methods typically require over 10 MLLM forward passes per image, whereas ACTIVE-O3 performs parallel region selection in a single forward pass.

### 5.2 OPEN-WORLD SMALL/DENSE OBJECT GROUNDING

**Dataset.** We build our benchmark on the LVIS dataset (Gupta et al., 2019), known for its rich long-tail vocabulary and abundance of small, densely packed objects. For small object grounding, we use instances under 100 pixels; for dense grounding, we select images with over 15 annotated instances. In both cases, we replace `<object>` in instruction $\mathcal{I}_O$ with the target category. We sample 10,000 training images and 1,200 validation images, ensuring each category appears no more than three times in the test set for balance. For all dataset details, please refers to Appendix E.3.

Table 1: Comparison of grounding and detection performance on **LVIS$_{small}$** and **LVIS$_{dense}$**. Numbers in parentheses denote improvements over the corresponding baseline.

| Method | LVIS$_{small}$ | | LVIS$_{dense}$ | | | | | |
| | AP$_s$ | AR$_s$ | AP$_s$ | AR$_s$ | AP$_m$ | AR$_m$ | AP$_l$ | AR$_l$ |
|---|---|---|---|---|---|---|---|---|
| Qwen2.5-VL | 1.2 | 1.8 | 1.6 | 2.0 | 9.7 | 11.0 | 15.0 | 18.7 |
| GDINO | 0.5 | 1.2 | 5.7 | 6.3 | 20.2 | 22.5 | 40.2 | 44.9 |
| Qwen2.5-VL-CoT | 1.2 | 2.2 | 2.5 | 3.5 | 11.2 | 14.4 | 20.3 | 25.8 |
| V* + GDINO | 0.6 | 1.4 | 5.7 | 6.7 | 18.3 | 22.9 | 32.8 | 42.6 |
| ACTIVE-O3 | 2.2 (+1.0) | 4.6 (+2.8) | 4.3 (+2.7) | 5.5 (+3.5) | 14.3 (+4.6) | 19.7 (+8.7) | 20.9 (+5.9) | 33.3 (+14.6) |
| ACTIVE-O3+GDINO | 1.2 (+0.7) | 2.5 (+1.3) | 7.0 (+1.3) | 7.9 (+1.6) | 25.1 (+4.9) | 29.3 (+6.8) | 45.1 (+4.9) | 55.9 (+11.0) |

**Results.** This benchmark is challenging due to small, densely packed objects. As shown in Table 1, both GDINO and Qwen2.5-VL struggle in this setting. In contrast, **ACTIVE-O3** outperforms Qwen2.5-VL and its CoT variant, improving AP$s$/AR$s$ by +1.0/+2.8 on **LVISsmall**, and by +2.7/+3.5 on **LVISdense**. It also improves AR$_l$ by +14.6 in large-object retrieval. When paired with GDINO, **ACTIVE-O3**+GDINO achieves 7.0 AP$_s$ and 7.9 AR$_s$, surpassing GDINO by +1.3/+1.6. These results highlight ACTIVE-O3 as a strong and generalizable sensing policy $\mathcal{M}_O$ for complex, open-world scenarios.

## 5.3 DOMAIN-SPECIFIC SMALL OBJECT DETECTION

**Dataset.** To evaluate domain generalization, we use the SODA benchmark (Cheng et al., 2023), which includes two large-scale datasets for small object detection: SODA-D (autonomous driving) and SODA-A (aerial imagery). SODA-D has 24,828 traffic images with 278,433 instances in 9 categories, while SODA-A offers 2,513 aerial images with 872,069 instances across 9 classes like vehicles and buildings. These datasets cover diverse and practical small-object detection scenarios.

**Results.** Table 2 shows that **ACTIVE-O3** achieves strong performance across both domains, with 9.2/10.4 AP$_s$/AR$_s$ on **SODA-A** and 15.1/22.0 on **SODA-D**. Despite the larger domain gap in the aerial scenario, **ACTIVE-O3** still outperforms Qwen2.5-VL by +8.5 AP$_s$ on SODA-A, indicating robust generalization. Performance on SODA-D is even higher,

Table 2: Performance comparison on **SODA-A** and **SODA-D** for small object detection. Numbers in parentheses denote improvement over Qwen2.5-VL.

| Method | SODA-A | | SODA-D | |
| | AP$_s$ | AR$_s$ | AP$_s$ | AR$_s$ |
|---|---|---|---|---|
| Qwen2.5-VL | 0.7 | 1.5 | 2.1 | 4.5 |
| GDINO | 0.5 | 1.2 | 8.0 | 8.7 |
| Qwen2.5-VL-CoT | 3.2 | 5.2 | 7.8 | 15.2 |
| ACTIVE-O3 | 9.2 (+8.5) | 10.4 (+8.9) | 15.1 (+13.0) | 22.0 (+17.5) |
| ACTIVE-O3+GDINO | 2.9 (+2.2) | 2.8 (+1.3) | 20.2 (+12.2) | 24.7 (+16.2) |

suggesting that our learned sensing policy $\mathcal{M}_O$ effectively transfers across distinct visual domains.

## 5.4 EFFECT OF RL TRAINING ON GENERAL VISUAL GROUNDING.

Although our framework is designed for active perception in challenging settings (e.g., small objects and dense scenes), we further evaluate whether the learned sensing policy also benefits standard visual grounding tasks.

We test ACTIVE-O3 on RefCOCO, RefCOCO+, and RefCOCOg using the same inference settings as Qwen2.5-VL.

Table 3: Accuracy (%) on RefCOCO/+/g visual grounding benchmarks.

| Model | RefCOCO | RefCOCO+ | RefCOCOg |
|---|---|---|---|
| Qwen2.5-VL | 88.7 | 83.1 | 86.2 |
| ACTIVE-O3 | **89.7** | **84.2** | **86.5** |

These results suggest that the active sensing policy learned in ACTIVE-O3 can also provide gains on general visual grounding tasks, even without being specifically optimized for them.

## 5.5 FINE-GRAINED INTERACTIVE SEGMENTATION

**Dataset and Setup.** We use the ThinObjects dataset for its fine-grained segmentation masks and semantic labels, ideal for evaluating zoom-in interactive segmentation. Due to the lack of a strong public task model $\mathcal{M}_A$, we use an oracle version that simulates perfect click-based feedback to isolate the impact of our sensing policy $\mathcal{M}_O$. Each sample allows up to 3 zoom-in steps, and performance is measured by mean IoU between predicted and ground-truth masks after interaction.

**Effect of Zoom-in Budget.** Figure 3 compares QWEN2.5-VL-COT and ACTIVE-O3 under different zoom-in budgets. While both start at the same initial mIoU, QWEN2.5-VL-COT suffers performance degradation as budget increases, dropping to 0.561 at budget 3. This is due to its tendency to zoom into incorrect regions, compounding errors in subsequent steps. In contrast, ACTIVE-O3 progressively improves to 0.863, demonstrating that our reinforcement learning policy effectively learns to identify and correct errors by selectively zooming in on challenging regions.

## 5.6 EFFECT OF RL TRAINING ON GENERAL VISUAL UNDERSTANDING

**Generalization to Reasoning and QA.** Although ACTIVE-O3 is not explicitly trained on reasoning or QA data, we observe stable or improved performance over its initialization model (Qwen2.5-VL-7B-Instruct) on a range of standard benchmarks, including MMBench (Liu et al., 2024c), MME (Yin et al., 2024), and RealWorldQA (xAI, 2024) (Table 4). Importantly, no benchmark shows degradation, indicating that our RL training enhances active perception capabilities without harming generalization. Moreover, on several benchmarks such as RealWorldQA, we ob-

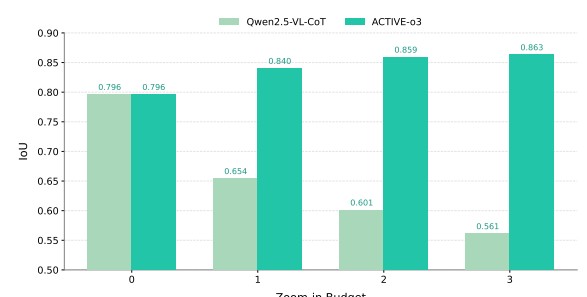

Figure 3: Comparison of segmentation performance (mIoU) under different zoom-in budgets.

serve clear improvements, suggesting that active perception can function as an effective proxy objective: by leveraging perception annotations, it not only strengthens active perception capability but also indirectly improves the model's visual understanding and reasoning capacity.

Table 4: Effect of RL training on general visual understanding. Results are reported on MMBench (`dev_en_v11`), MME, and RealWorldQA. ACTIVE-O3 consistently maintains or improves performance over its initialization model.

| Method | MMBench | MME | RealWorldQA |
|---|---|---|---|
| Qwen2.5-VL-7B-Instruct | 80.1 | 2308 | 67.9 |
| ACTIVE-O3 | 80.5 | 2316 | 69.7 |

## 6 CONCLUSION

We propose **ACTIVE-O3**, a pure reinforcement learning framework that empowers MLLMs with active perception via a two-module policy for sensing and action. Trained with task-aware and exploratory rewards, ACTIVE-O3 enables MLLMs to reason about where to look and how to act more effectively. Experiments across open-world grounding, fine-grained segmentation, and domain-specific small object detection show that ACTIVE-O3 consistently improves accuracy and efficiency under limited computational budgets, while generalizing well across diverse domains. We hope that this work encourages further research on active vision with MLLMs.

ETHICS STATEMENT

This work investigates reinforcement learning for active perception in multimodal large language models (MLLMs). The research is conducted entirely in controlled academic settings using public datasets, without involving sensitive or private data. Potential concerns such as dataset bias or over-reliance of agents on imperfect perception may exist in general for vision–language systems, but our work does not introduce new risks beyond those already present in existing MLLMs. We encourage future deployments to follow standard guidelines on fairness, transparency, and responsible use. Overall, we believe our framework brings positive impact by improving efficiency, interpretability, and safety of active perception in AI systems.

**Risks, Misuse, and Mitigations.** - *Privacy and Surveillance.* If deployed in camera-equipped systems, enhanced perception may be misused for surveillance or privacy-intrusive tasks beyond intended scope. To mitigate this, any deployment should follow local regulations, anonymize or blur sensitive regions (e.g., faces), and strictly limit allowable use cases. - *Bias and Fairness.* Visual perception systems often inherit dataset biases (e.g., underrepresentation of certain object classes, environmental or demographic biases). We encourage evaluating on diverse datasets and reporting failure cases or disparities. - *Overreliance / Misbehavior.* Agents might overconfidently act based on imperfect perception proposals. We recommend fallback mechanisms, human oversight, or uncertainty estimation modules in real-world deployment.

REPRODUCIBILITY STATEMENT

We are committed to ensuring reproducibility of our results. All implementation details, hyperparameters, model configurations, and evaluation protocols are provided in the main paper and appendix.

- **Code and Data.** We will release the full source code, training scripts, evaluation scripts, and benchmark datasets upon paper acceptance.
- **Hyperparameters and Setup.** In Appendix E.2, we provide full hyperparameter (learning rates, batch sizes, number of steps, exploration/noise parameters, model architecture details, etc.).
- **Random Seeds and Runs.** We conduct multiple independent runs (e.g. 3 seeds) and report mean and standard deviation for all performance metrics in Appendix F.
- **Pretrained Models.** We specify the pretrained MLLM model used (e.g. Qwen2.5-VL-7B) and any weights or checkpoints we fine-tune or freeze.
- **Evaluation Protocol.** All experimental settings are aligned across compared methods to ensure fairness. For our proposed benchmark, we will release the dataset and evaluation suite upon acceptance, with implementation details provided in Appendix E.3. For other public benchmarks, we adopt the official evaluation scripts provided by VLMevalKit.
- **Ablation Analyses.** We include ablation studies (e.g. on the reward design, selection of regions, different dataset combinations) in Appendix G, to show robustness to design choices.

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

## USE OF LLMs

We used large language models (LLMs) only for minor assistance in polishing the language and adjusting the presentation of tables. No LLMs were involved in designing the methodology, conducting experiments, or analyzing results.

## A    APPENDIX OVERVIEW

This appendix provides additional technical details, implementation insights, and extended results to supplement the main paper. It is organized as follows:

- **Section B: Heuristic Reward Formulations**
  Describes the manually designed reward components used to evaluate MLLM outputs, including format validity, spatial overlap, area constraints, and coverage metrics.
- **Section C: Task-Aware Reward Formulation**
  Defines the reward signals computed using downstream task-specific models (e.g., object detection and interactive segmentation).
- **Section D: Discussion: Framework Considerations and Insights**
  Discusses the design choices and considerations behind our MLLM-based active perception framework.
- **Section E: Method Details**
  Discusses implementation details of our active perception system, including MLLM prompt design, reward integration, evaluation metrics, and model configuration.
- **Section G: Ablation Studies**
  Presents ablation experiments on different reward combinations and dataset configurations to understand the contribution of each component.
- **Section H: Qualitative Visualization**
  Visual comparisons of model outputs, including correct cases and failure modes, to highlight model behavior under different conditions.

## B    HEURISTIC REWARD FORMULATIONS

In this section, we detail the heuristic reward functions used to evaluate the quality of region proposals generated by the MLLM. Each reward component is applied to a single MLLM response $y$, which typically includes multiple bounding boxes $\{b_i\}_{i=1}^N$ and optional reasoning traces. The final reward $\mathcal{R}_{\text{heuristic}}(y)$ is a weighted combination of the components described below.

### B.1    FORMAT VALIDITY REWARD $\mathcal{R}_{\text{FORMAT}}$

This reward ensures the response adheres to expected syntax and structure. It includes two checks:

- **JSON validity:** the output must be parseable as a list of objects with bounding box fields `bbox_2d`.
- **Response structure:** the output should include the required reasoning and answer format using tags `<think>` and `<answer>`.

$$\mathcal{R}_{\text{format}}(y) = \begin{cases} 1, & \text{if } y \text{ is valid JSON and contains both } \texttt{<think>}, \texttt{<answer>} \\ 0, & \text{otherwise} \end{cases}$$

### B.2    NON-OVERLAPPING REWARD $\mathcal{R}_{\text{NO-OVERLAP}}$

This reward penalizes overlapping region proposals to promote spatial diversity:

$$\mathcal{R}_{\text{no-overlap}}(\{b_i\}) = \begin{cases} 1, & \text{if } \text{IoU}(b_i, b_j) \leq \tau, \ \forall i \neq j \\ 0, & \text{otherwise} \end{cases} \quad \text{with } \tau = 0.3$$

### B.3 AREA RANGE REWARD $\mathcal{R}_{\text{AREA}}$

We encourage region proposals whose areas fall within a reasonable proportion of the image:

$$\text{AreaRatio}(b_i) = \frac{(x_2 - x_1 + 1)(y_2 - y_1 + 1)}{W \cdot H}$$

$$\mathcal{R}_{\text{area}}(\{b_i\}) = \begin{cases} 1, & \text{if } \forall i, \ r_{\min} \leq \text{AreaRatio}(b_i) \leq r_{\max} \\ 0, & \text{otherwise} \end{cases} \quad \text{with } r_{\min} = 0.01, \ r_{\max} = 0.5$$

### B.4 COVERAGE-BASED REWARD $\mathcal{R}_{\text{COVERAGE}}$

This reward evaluates how well the proposed regions align with task-relevant areas. It is defined in multiple modes:

- **Ground-truth mask coverage:** for binary mask $M \in \{0, 1\}^{H \times W}$, we compute the average proportion of mask pixels covered:

$$\text{Coverage}(b_i, M) = \frac{\sum_{(x,y) \in b_i} M(x, y)}{\text{Area}(b_i)}$$

$$\mathcal{R}_{\text{mask}}(\{b_i\}) = \frac{1}{N} \sum_{i=1}^{N} \mathbf{1}\left[\text{Coverage}(b_i, M) \geq \theta\right]$$

- **Ground-truth box coverage:** we count how many ground-truth boxes have at least one matching predicted box (IoU $\geq \delta$), producing a coverage ratio:

$$\mathcal{R}_{\text{gt-box}} = \frac{\#\text{matched GT boxes}}{\#\text{total GT boxes}}$$

- **Mask-to-mask alignment:** if both predicted and ground-truth masks are available, we compute Dice or IoU over the merged regions.

The final coverage reward can be defined as a soft combination of the above modes when applicable.

### B.5 OVERALL HEURISTIC REWARD

We define the total heuristic reward as a weighted sum of the components:

$$\mathcal{R}_{\text{heuristic}}(y) = \lambda_1 \mathcal{R}_{\text{format}} + \lambda_2 \mathcal{R}_{\text{no-overlap}} + \lambda_3 \mathcal{R}_{\text{area}} + \lambda_4 \mathcal{R}_{\text{coverage}}$$

where $\lambda_i$ are all set to 1.

## C TASK-AWARE REWARD FORMULATION

We provide task-specific definitions of the reward signal computed from the outputs of the task model $\mathcal{M}_A$.

**Object Detection.** Let $\hat{B} = \{b_i\}_{i=1}^{K}$ be the predicted bounding boxes and $B^* = \{b_j\}_{j=1}^{J}$ be the ground-truth boxes. The reward is computed using standard detection metrics:

$$\mathcal{R}_{\text{detect}} = \text{AP@IoU=0.5} + \text{AR@IoU=0.5}$$

**Interactive Segmentation.** Let $\hat{M}$ be the predicted mask returned by the SAM (Ravi et al., 2024) API and $M^*$ be the ground-truth mask. The segmentation reward is defined as:

$$\mathcal{R}_{\text{seg}} = \text{mIoU}(\hat{M}, M^*) = \frac{|\hat{M} \cap M^*|}{|\hat{M} \cup M^*|}$$

We generate the SAM prediction using positive and negative points inferred by $\mathcal{M}_A$.

# D    DISCUSSION: FRAMEWORK CONSIDERATIONS AND INSIGHTS

In this section, we provide further insights into the design of our MLLM-based active perception framework, building upon the main formulation introduced in Section 3 of main paper. The following remarks highlight critical architectural choices and theoretical simplifications made to improve performance, efficiency, and generalization.

*Remark* D.1 (**MLLM-Driven Action and Sensing Modules**). Unlike prior approaches that use specialist models for each module, we adopt a single multi-modal large language model (MLLM) to jointly handle both action and sensing. This design offers several advantages. First, MLLMs exhibit strong capabilities in following natural language instructions and generalizing to open-ended semantic goals. Second, they can leverage rich contextual information, including long-term observation history, to make more informed and coherent decisions. Finally, in addition to predicting $a_t^{\text{env}}$ and $a_t^{\text{cam}}$, MLLMs can also generate intermediate reasoning steps, which not only enhance interpretability but have also been shown to improve task performance in prior work (e.g., chain-of-thought prompting).

*Remark* D.2 (**Optimization Strategy**). In principle, the action model $\mathcal{M}_A$ and the sensing model $\mathcal{M}_O$ can be jointly optimized. However, this requires $\mathcal{M}_A$ to already possess sufficient baseline capability. A common alternative is to perform staged or iterative optimization, where one alternately updates $\mathcal{M}_A$ and $\mathcal{M}_O$ in a bootstrapping manner. In this work, we assume access to a reasonably strong $\mathcal{M}_A$ and focus on optimizing $\mathcal{M}_O$ accordingly, since our goal is to investigate how to equip MLLMs with effective active perception strategies. Furthermore, to simplify the problem, we reformulate the perceptual cost term as a fixed sensing budget. That is, under a given number of allowed sensing actions, the objective becomes maximizing task reward. This is the setup we adopt in our experiments.

*Remark* D.3 (**Specialization to 2D Visual Scenesy**). While our general formulation applies to embodied agents in complex physical environments, such settings are often difficult to deploy and evaluate in a reproducible manner. To facilitate more controlled and fair comparisons, we specialize the problem to a simplified yet representative 2D scenario: active perception over static images. In this setting, the environment state $s_t^{\text{env}}$ is a high-resolution static image $I \in \mathbb{R}^{H \times W \times 3}$. The sensing action $a_t^{\text{cam}}$ specifies a rectangular region within $I$, parameterized as a bounding box $(x, y, w, h)$ [1]. The observation $o_t$ is obtained by cropping the region defined by $a_t^{\text{cam}}$ from $I$ and resizing it to a fixed resolution :

$$o_t = \text{ResizeCrop}(I, \ a_t^{\text{cam}})$$

The task model $\mathcal{M}_A$ then operates on the selected region to perform downstream functions such as classification, detection, or answering visual questions. This setting preserves the core challenge of active perception—selecting informative views—while simplifying execution and enabling systematic evaluation.

*Remark* D.4 (**2D Setting as a Single-Step Active Perception Problem**). A key property of the 2D visual scenario is that the environment state $s^{\text{env}}$ remains static across time (since the interaction action $a_t^{\text{env}}$ does not change the image). As a result, the task reduces to a single-step decision problem ($T = 1$), and the agent's objective becomes repeatedly selecting an initial sensing action $a_0^{\text{cam}}$. This reframing allows for a significantly more efficient implementation: multiple candidate sensing actions can be evaluated in parallel, enabling broader exploration of the observation space without relying on sequential interaction. In this sense, 2D active perception can be viewed as a parallelized search over viewpoints within a fixed scene.

*Remark* D.5 (**GPT-o3 vs. ACTIVE-O3**). The zoom-in search strategy used in GPT-o3 can be seen as a special case of the active perception framework defined in this paper. However, it suffers from two major limitations. First, its search process is purely sequential—only one region can be selected and zoomed in at a time—which leads to low efficiency. Second, its region selection is often inaccurate, resulting in unnecessary zooms and missed critical areas. In contrast, ACTIVE-O3 enables parallel selection of multiple candidate regions, improving search coverage and efficiency. Moreover, by leveraging the reasoning capability of MLLMs and optimizing the sensing policy $\mathcal{M}_O$ through reinforcement learning, ACTIVE-O3 is able to identify more informative regions under a fixed sensing budget.

---

[1]We focus on axis-aligned rectangular regions and omit rotation for simplicity, although it can be incorporated into the action space.

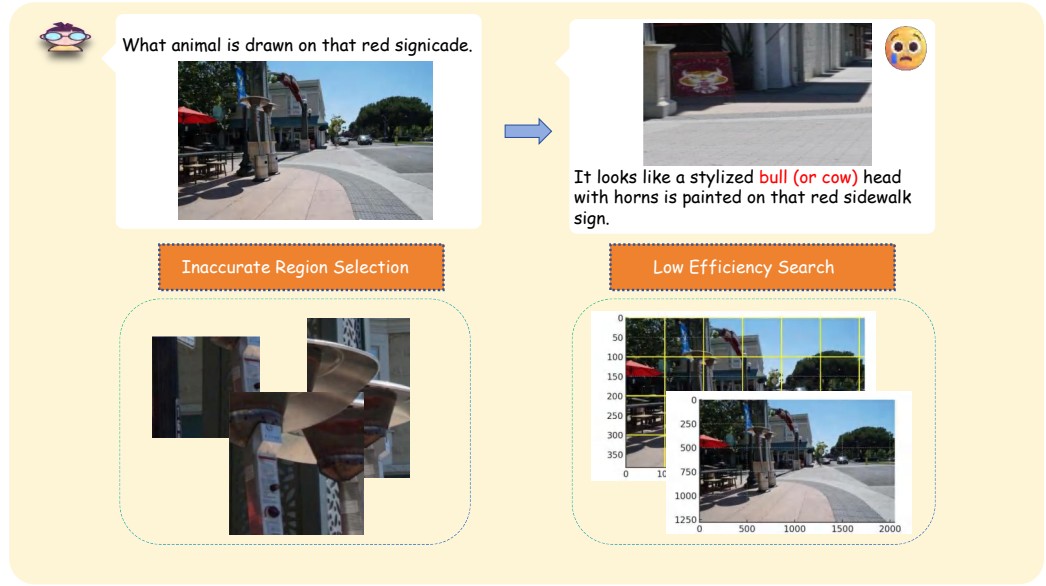

Figure 4: A failure case of GPT-o3 in answering the question: What animal is drawn on that red signicade?. The reasoning trajectory reveals two key limitations: inaccurate region selection (left), and inefficient, near-exhaustive search patterns (right).

We further illustrate the limitations of GPT-o3 with a failure case shown in Figure 4. The task is to answer the question "What animal is drawn on that red signicade?". The correct answer is tiger, as a stylized tiger face is clearly visible on the red sidewalk sign.

However, GPT-o3 fails to accurately locate the relevant region. It initially zooms into irrelevant parts of the image—such as metallic structures and background textures—due to its limited context and short-horizon planning. As shown in the left panel of Figure 4, the chosen regions completely miss the actual sign.

Moreover, as highlighted on the right side, GPT-o3's sensing process becomes inefficient, closely resembling exhaustive grid-based search in some cases. This leads to redundant actions and poor use of the limited zoom-in budget. In contrast, ACTIVE-O3 identifies more informative regions early by reasoning over spatial layout and task context, significantly improving efficiency and accuracy.

### D.1 LIMITATIONS AND FUTURE WORK

Despite the promising results, our framework has several limitations that open avenues for future research. (see Figure 18).

First, the domain gap remains a challenge, particularly for specialized domains such as remote sensing. Current MLLMs may struggle to accurately identify domain-specific categories (e.g., windmills, storage tanks), which can lead to inaccurate task-aware reward estimation due to the limited capability of the task model.

Second, the current action space is constrained. Our framework only allows zooming into three target regions per step. However, certain applications may require more flexible control, such as selecting a larger number of regions or introducing transformations like rotation—especially relevant for tasks like OCR, though less critical for tasks such as grounding.

Third, the input to the sensing model is limited to the current observation. In practice, incorporating a memory mechanism to store past actions and observations could enable more informed decision-making. This extension may support more sophisticated strategies, such as trajectory-level planning, long-term search, and rollback operations.

> ## Prompt for ACTIVE-O3 Detection
>
> - "Find up to three different regions in the image that likely contain a high number of '**{object}**'."
> - "Even if the '**{object}**' are not clearly visible, infer where they are most likely to appear."
> - "Each region should cover multiple '**{object}**' and include some visual context."
> - "The selected regions should be as distinct as possible, with minimal or no overlap between them."
> - "Return the coordinates in JSON format as: {"bbox_2d": [x1, y1, x2, y2], "label": "**{object}**-dense region"}."
> - "Explain your reasoning in `<think>`...`</think>` and output the final result in `<answer>`...`</answer>`."
> - "Example: `<think>` thinking process here `</think>` `<answer>` JSON format here `</answer>`"

Figure 5: Prompt for ACTIVE-O3-DET.

> ## Prompt for ACTIVE-O3 Segmentation
>
> - "Identify exactly three distinct regions in the image that illustrate segmentation inaccuracies in the translucent green mask for the '**{object}**'."
> - "The selected regions should be as distinct as possible, with minimal or no overlap between them."
> - "Check whether the mask accurately covers the '**{object}**', meaning it should fully include the object without significant over-segmentation (mask extends into background) or under-segmentation (parts of the object are not covered)."
> - "Each region should represent a clear segmentation mistake and include enough surrounding context for verification."
> - "Return the results in JSON format as: {"bbox_2d": [x1, y1, x2, y2], "label": "**{object}** segmentation error"}."
> - "Explain your reasoning in `<think>`...`</think>` and output the final result in `<answer>`...`</answer>`."
> - "Example: `<think>` reasoning process here `</think>` `<answer>` JSON format here `</answer>`"

Figure 6: Prompt for ACTIVE-O3-Seg.

Addressing these limitations could further improve the adaptability, generalization, and decision quality of the proposed sensing policy in more complex or specialized scenarios.

## E  METHOD DETAILS

### E.1  PROMPT DESIGN

In this section, we provide the prompts used to guide the MLLM in both detection (Figure 2) and segmentation (Figure 6) tasks as the sensing policy $\mathcal{M}_O$. The prompts are designed to elicit specific behaviors from the model, ensuring that it generates appropriate region proposals and reasoning. For the task model $\mathcal{M}_A$, we use an simple instruction to ask the model to perform the task (Figure 7).

### E.2  IMPLEMENTATION DETAILS

We use Qwen2.5-VL-7B-Instruct as the shared policy backbone $\pi_\theta$. All experiments are conducted using GRPO with KL regularization coefficient $\beta = 0.04$, group size 8, and a learning rate of $1e{-}6$ using the AdamW optimizer with weight decay $0.01$.

Training is performed on 8 GPUs with 80–90GB memory each, using bf16 precision, a per-device batch size of 1, gradient accumulation of 1, and gradient checkpointing enabled. Training is performed with DeepSpeed ZeRO-3 for memory efficiency. Each experiment typically completes within 24 hours. For the sensing model $\mathcal{M}_O$, we resize the input image such that the shorter side is

---

**Prompt for Task Model**

- "Please find all instances of '**{object}**' in the image and return the bounding box coordinates in JSON format."

---

Figure 7: Prompt for the task model $\mathcal{M}_A$.

1024 pixels, while preserving the original aspect ratio. For the task model $\mathcal{M}_A$, all images are resized to a fixed resolution of $840 \times 840$. For Grounding DINO, we follow the official preprocessing pipeline provided by the authors.

### E.3 DATASETS DETAILS

**LVIS.** We construct our benchmark for open-world small and dense object grounding based on the LVIS (Gupta et al., 2019) dataset, which offers the richest long-tail object vocabulary and the highest prevalence of small and densely packed instances among existing segmentation datasets. To assess small object grounding, we identify all instances with an area less than 100 pixels and retain their corresponding categories as test queries. For dense object grounding, we select images that contain more than 15 annotated instances and treat all instance categories within such images as query targets. In both cases, we replace the placeholder <object> in the original instruction $\mathcal{I}_O$ with the chosen category name. We sample 10,000 training images from the LVIS training set using this strategy, and 1,200 images from the validation set for evaluation. During test set construction, we ensure that each category appears at most three times to promote category balance. We adopt standard COCO evaluation metrics using the official COCO API. Specifically, we report average precision (AP) across IoU thresholds from 0.5 to 0.95 (in 0.05 increments), as well as AP for small ($AP_s$), medium ($AP_m$), and large ($AP_l$) object sizes.

**SODA.** To further evaluate the generalization of our framework in specialized visual domains, we adopt the SODA (Cheng et al., 2023) benchmark, which includes two large-scale datasets designed for small object detection: **SODA-D** (autonomous driving) and **SODA-A** (aerial imagery). SODA-D contains 24,828 traffic images with 278,433 annotated instances across nine traffic-related categories. SODA-A includes 2,513 high-resolution aerial images with 872,069 object instances across nine categories such as vehicles and buildings. These datasets present a wide range of realistic and challenging small-object detection scenarios. During training, we randomly select 1,000 images from each dataset as the training set. For SODA-A, whose annotations are originally provided as polygons, we convert them into bounding boxes to serve as ground truth for training and evaluation. Due to the significant domain shift compared to LVIS, direct use of standard evaluation settings (e.g., COCO-style AP at IoU 0.5–0.95) leads to very low scores and poor comparability. To better capture performance under such domain-specific conditions, we lower the IoU threshold to 0.1 when computing detection metrics. This adjustment allows a fairer evaluation of the model's generalization ability in these more challenging domains.

**ThinObjects.** Following SegAgent(Zhu et al., 2025), we adopt the ThinObjects (Liew et al., 2021) dataset for this task, as it provides both semantic annotations and high-quality, fine-grained segmentation masks, making it suitable for evaluating interactive segmentation under zoom-in conditions. One core challenge is the lack of a robust existing task model $\mathcal{M}_A$ for click-based interactive segmentation. To focus on evaluating the effectiveness of our method as a sensing policy $\mathcal{M}_O$, we construct an oracle variant of $\mathcal{M}_A$ as a proxy. This oracle simulates perfect feedback during interaction. We set a maximum budget of 3 zoom-in steps per sample. The final performance is evaluated using the mean Intersection over Union (mIoU) between the predicted and ground-truth masks after the interaction sequence.

## F  ROBUSTNESS TO RANDOM SEEDS

To further investigate the influence on training stability brought about by our compositional reward design, we conducted additional experiments to evaluate the robustness of our method on different random seeds.

**Experiments**  We train our Active-o3 model using three different random seeds (0, 1, and 2) while keeping all other hyperparameters and training configurations identical. Each model is trained for the same number of iterations and evaluated on the same test set. The low standard deviations across different training seeds demonstrate the training stability of our compositional reward design.

Table 5: Performance using difference training random seeds

| Run | $AP_s$ | $AR_s$ | $AP_m$ | $AR_m$ | $AP_l$ | $AR_l$ |
|---|---|---|---|---|---|---|
| Active o3 | 4.3 | 5.5 | 14.3 | 19.7 | 20.9 | 33.3 |
| seed 0 | 4.3 | 5.5 | 14.0 | 18.9 | 22.1 | 33.7 |
| seed 1 | 4.3 | 5.6 | 14.5 | 20.4 | 20.6 | 31.6 |
| seed 2 | 4.1 | 5.4 | 13.6 | 18.9 | 17.8 | 29.8 |
| Avg $\pm$ Std | $4.25 \pm 0.10$ | $5.50 \pm 0.08$ | $14.10 \pm 0.39$ | $19.48 \pm 0.72$ | $20.35 \pm 1.82$ | $32.10 \pm 1.78$ |

## G  ABLATION STUDIES

### G.1  ANALYSIS OF DIFFERENT RL ALGORITHMS

To verify the generalizability of our proposed framework and explore the potential of different optimization strategies, we compared GRPO with other mainstream RL algorithms, including PPO, Reinforce++, and two recent GRPO variants: GMPO and GPG. The quantitative results are summarized in Table 6.

Table 6: Performance comparison of different RL algorithms on the active perception task.

| Method | $AP_s$ | $AR_s$ | $AP_m$ | $AR_m$ | $AP_l$ | $AR_l$ |
|---|---|---|---|---|---|---|
| Reinforce++ | 2.73 | 3.34 | 12.47 | 19.12 | 13.15 | 25.31 |
| PPO | 4.01 | 5.06 | 15.07 | 19.73 | 17.04 | 26.94 |
| GMPO | **4.46** | **5.72** | 13.58 | 19.13 | 17.63 | 27.32 |
| GPG | 3.83 | 5.13 | **15.07** | **21.02** | 19.35 | **31.41** |
| **GRPO (Ours)** | 4.06 | 5.32 | 13.96 | 19.10 | **20.63** | 30.05 |

**Result Analysis.**  The results demonstrate that our framework is robust across various RL optimizers. **Reinforce++** exhibits the lowest performance due to high variance, while **GRPO** provides a strong balance between stability and precision ($AP_l$ 20.63). Notably, recent variants like **GMPO** (best on small objects) and **GPG** (best recall on large objects) show distinct strengths. It is important to acknowledge that these results were obtained under unified settings aligned with GRPO; considering that optimal hyperparameters vary across algorithms, methods like GPG might achieve even higher performance with extensive specific tuning. Overall, these diverse results highlight the **broad prospects** for future research. The capability of different algorithms to optimize specific metrics suggests that developing specialized RL strategies for MLLM-based active perception is a promising avenue to further enhance model capabilities.

### G.2  COMPARISON WITH DIRECT REINFORCEMENT FINE-TUNING

To verify that our performance gains stem from the active perception mechanism rather than merely applying reinforcement learning to the VLM, we compared our method with a baseline named **"Direct RFT"**. In Direct RFT, we directly fine-tune the task model using GRPO on the same training data, but without the sensing module (i.e., the model processes the original image directly).

Table 7: Performance comparison between Active-o3 and Direct RFT on LVIS and SODA datasets.

| Method | LVIS$_{dense}$ | | | | | | SODA-A | | SODA-D | |
| | $AP_s$ | $AR_s$ | $AP_m$ | $AR_m$ | $AP_l$ | $AR_l$ | $AP_s$ | $AR_s$ | $AP_s$ | $AR_s$ |
| --- | --- | --- | --- | --- | --- | --- | --- | --- | --- | --- |
| Direct RFT | 2.9 | 3.9 | 13.3 | **20.0** | **23.4** | **35.6** | 5.2 | 6.7 | 9.1 | 16.2 |
| **Active-o3** | **4.3** | **5.5** | **14.3** | 19.7 | 20.9 | 33.3 | **9.2** | **10.4** | **15.1** | **22.0** |

**Analysis.** The results in Table 7 reveal a clear distinction in capabilities. While Direct RFT achieves competitive performance on large objects ($AP_l$), it struggles significantly with small and dense objects. In contrast, **Active-o3** demonstrates a substantial improvement on small objects (e.g., +48% relative gain on LVIS $AP_s$, +77% on SODA-A $AP_s$, and +66% on SODA-D $AP_s$). This confirms that purely data-driven RL training (Direct RFT) is insufficient for scenarios requiring fine-grained visual details due to resolution limitations. The active sensing module is indispensable for bridging this gap, enabling the model to actively perceive and interpret challenging scene details.

G.3 COMPUTATIONAL COST ANALYSIS

To evaluate the efficiency of our proposed framework, we compared the computational cost of training Active-o3 against the Direct RFT baseline. We provide a detailed breakdown of the time consumption per training step in Table 8.

Table 8: Training time breakdown (in seconds) per step. "Without Grounding" represents the Direct RFT baseline, and "With Grounding" represents our Active-o3 method.

| Configuration | Generate Sequence | Transition | Old_logp | Update _actor | Ref | Grounding | **Total** |
| --- | --- | --- | --- | --- | --- | --- | --- |
| Without Grounding | 2.5s | 5.7s | 2.3s | 7.2s | 2.9s | - | **20.6s** |
| With Grounding | 2.5s | 5.7s | 2.3s | 7.2s | 2.9s | 3.5s | **24.1s** |

**Detailed Stage Definitions.**

- **Generate Sequence:** Model inference time to generate group sequences.
- **Transition:** Time for syncing actor weights to the inference engine and launching the rollout process.
- **Old_logp:** Computing log probabilities of actions under the old policy.
- **Update_actor:** Gradient computation and parameter updates for the policy network.
- **Ref:** Computing reference model outputs for GRPO importance ratio and KL regularization.
- **Grounding:** Execution time of the visual grounding module to compute the Task Reward.

**Analysis.** As shown in Table 8, the computational cost for the fundamental RL processes remains consistent across both configurations. The introduction of the Task Reward in Active-o3 requires an additional inference pass by the visual grounding module, adding only 3.5 seconds per step. This leads to a marginal increase in total training time of approximately 17%. Considering the significant performance gains achieved in small and dense object perception, we conclude that Active-o3 offers a highly efficient trade-off between computational cost and model performance.

**Effect of candidate region quantity.** To validate the benefits of multi-candidate selection for performance improvement, we systematically vary the number of selected regions from 1 to 3. Table 9 presents the results across different object scales. The performance demonstrates a consistent upward trend as the number of selected regions increases, indicating that our method maintains high effectiveness in each selection round. Notably, each additional candidate region contributes meaningfully to the overall performance gain, demonstrating our method's ability to consistently identify valuable regions without redundant selections.

Table 9: Ablation study on different number of selected regions

| Boxes | $AP_s$ | $AR_s$ | $AP_m$ | $AR_m$ | $AP_l$ | $AR_l$ |
|-------|--------|--------|--------|--------|--------|--------|
| 1 | 2.98 | 3.12 | 10.34 | 11.31 | 21.38 | 23.67 |
| 2 | 4.24 | 4.66 | 16.18 | 18.59 | 27.19 | 32.92 |
| 3 | 4.86 | 5.52 | 19.23 | 22.93 | 36.03 | 44.28 |

**Parallel selection vs. repeated sampling.** To demonstrate that Active-o3 performs effective target region selection rather than benefiting merely from multiple attempts, we compare our parallel joint selection strategy against repeated single region sampling. Table 10 shows the comparison between our approach and a baseline that samples 1 region at a time but repeats for 3 times using the same selection policy model. Our method achieves superior performance across all object scales with notable improvements. These results highlight the effectiveness of our joint selection mechanism, which considers multiple candidate regions simultaneously to achieve optimal spatial coverage.

Table 10: Comparison of different selection strategy

| Method | $AP_s$ | $AR_s$ | $AP_m$ | $AR_m$ | $AP_l$ | $AR_l$ |
|--------|--------|--------|--------|--------|--------|--------|
| Active o3 + GDINO | 7.0 | 7.9 | 25.1 | 29.3 | 45.1 | 55.9 |
| Sampling 3 times | 4.9 | 5.5 | 19.2 | 22.9 | 36.0 | 44.2 |

**Training Data Combination.** Table 11 presents the effect of different training data combinations on small object detection performance, evaluated on SODA-A and SODA-D. When incorporating LVIS into the training set, the performance improves significantly across both domains. For example, adding LVIS to SODA-A yields a +2.7 $AP_s$ and +1.3 $AR_s$ gain on SODA-A, and also enables reasonable generalization to SODA-D. Finally, using the full combination of LVIS, SODA-A, and SODA-D leads to the best overall performance, achieving 9.2/10.4 on SODA-A and 15.1/22.0 on SODA-D. These results demonstrate that ACTIVE-O3 serves as a general and flexible framework capable of leveraging heterogeneous domain-specific datasets to learn a unified sensing policy $\mathcal{M}_O$. By incorporating diverse training sources such as LVIS, SODA-A, and SODA-D, ACTIVE-O3 is able to generalize effectively across multiple domains, highlighting its scalability and adaptability in open-world scenarios.

**Reward Design.** As mentioned in Section 4, we adopt a dual-form reward design that combines heuristic and task-aware rewards. To evaluate the impact of each component, we conduct an ablation study on the reward design. As shown in Table 12, the combined reward achieves the best performance across all object sizes, especially for small objects ($AP_s$: 4.4, $AR_s$: 5.8). Compared to using only task or heuristic rewards, the combination leads to consistent improvements, indicating that it effectively balances exploration (via heuristics) and task-driven optimization. This validates the effectiveness of our dual-form reward design in guiding better policy learning.

To verify whether our reward design preserves the general reasoning capabilities of the MLLM, we evaluated different reward configurations on three standard benchmarks. The results are shown in Table 13.

**Analysis.** The results indicate a **positive correlation between active perception capability and general visual understanding**. While the unguided *Heuristic-only* policy impairs reasoning (likely due to context loss from aggressive cropping), our full method ensures that active exploration remains semantically meaningful, thereby enhancing performance across all general benchmarks compared to the baseline.

Table 11: Impact of training data combinations on small object detection performance. We report $AP_s/AR_s$ on SODA-A and SODA-D.

| Training Set | SODA-A | | SODA-D | |
|---|---|---|---|---|
| | $AP_s$ | $AR_s$ | $AP_s$ | $AR_s$ |
| SODA-A | 3.7 | 7.5 | – | – |
| SODA-D | – | – | 11.4 | 18.9 |
| LVIS + SODA-A | 6.4 | 8.8 | 14.0 | 17.9 |
| LVIS + SODA-A + D | **9.2** | **10.4** | **15.1** | **22.0** |

Table 12: Ablation study on reward design. Comparison of task reward, heuristic reward, and their combination across different object sizes (small, medium, large). Metrics are AP and AR.

| Reward Type | $AP_s$ | $AR_s$ | $AP_m$ | $AR_m$ | $AP_l$ | $AR_l$ |
|---|---|---|---|---|---|---|
| Task Reward | 3.6 | 5.0 | 12.1 | 15.7 | 16.4 | 25.2 |
| Heuristic Reward | 3.0 | 4.2 | 9.7 | 13.8 | 13.2 | 21.7 |
| Combined Reward | **4.4** | **5.8** | **15.4** | **20.2** | **19.1** | **27.4** |

## H QUALITATIVE VISUALIZATION

### H.1 QUALITATIVE ANALYSIS OF REWARD COMPONENTS

To further investigate how different reward components influence the agent's active perception behavior, we present qualitative comparisons in Figure 8 and Figure 9.

**Role of Task Reward.** As shown in Figure 8, the Task Reward acts as a bridge between the sensing action and downstream recognition performance. Without it (Heuristic-only), the agent optimizes solely for geometric coverage, often resulting in sub-optimal inputs for the task model. For instance, in the *Flamingo* case, the heuristic baseline generates an excessively narrow bounding box. When resized for the task model, this severe aspect ratio distortion hinders recognition. In contrast, our Combined Reward drives the agent to decompose the target into three distinct, well-proportioned boxes. Similarly, in the *Hook* and *Bun* scenarios, the Task Reward encourages the preservation of semantic visual context (e.g., surrounding objects), which is critical for the task model to resolve ambiguities.

**Role of Heuristic Reward.** Figure 9 demonstrates the necessity of Heuristic Reward for efficiency. Relying solely on the Task Reward leads to inefficient exploration strategies characterized by high redundancy. In the *Motorcycle* and *Broccoli* examples, the Task-only baseline repeatedly samples overlapping regions. By incorporating the Heuristic Reward, the agent learns to avoid unnecessary overlap. This efficiency gain directly translates to better coverage; as seen in the *Garlic* scene, the agent avoids wasting steps on redundant views and successfully identifies a previously missed object in the top-right corner.

### H.2 ZERO-SHOT TRANSFER ON $V^*$ BENCHMARK

We demonstrate that ACTIVE-O3 is capable of zero-shot transfer to fine-grained VQA tasks, such as those in the $V^*$ (Wu & Xie, 2024) benchmark. By learning effective reasoning and search strategies through reinforcement learning on small object detection tasks, ACTIVE-O3 generalizes well to previously unseen tasks. We highlight several challenging cases involving OCR (Figures 11, 12) and attribute recognition (Figures 13, 14) where base models struggle. In contrast, ACTIVE-O3 can successfully complete the task by leveraging its ability to reason and zoom in adaptively.

### H.3 SMALL OBJECT DETECTION ON SODA-A AND SODA-D

Figure 15 presents qualitative results of ACTIVE-O3 on the SODA-A and SODA-D datasets. Compared with several baselines, ACTIVE-O3 consistently selects more relevant regions to zoom into, leading to improved detection performance on small objects. These results demonstrate that our

Table 13: Ablation study of reward configurations on general visual understanding benchmarks.

| Method | MMBench | MME | RealWorldQA |
|---|---|---|---|
| Qwen2.5-VL (Base) | 80.1 | 2308 | 67.9 |
| Active-o3 (Heuristic Reward only) | 78.9 | 2303 | 68.1 |
| Active-o3 (Task Reward only) | 80.0 | 2309 | 68.6 |
| **Active-o3 (Full)** | **80.5** | **2316** | **69.7** |

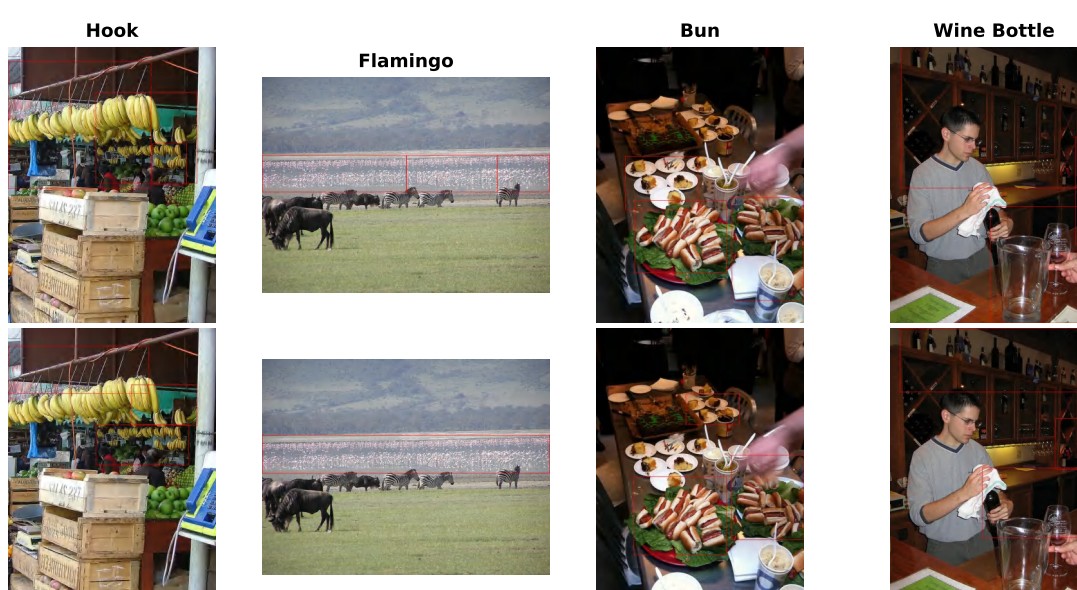

Figure 8: **Qualitative ablation of the Task Reward component.** We compare our full method (Combined Reward, top row) with a baseline using only Heuristic Reward (bottom row). Without the Task Reward, the sensing model focuses solely on object coverage, often ignoring visual context or producing extreme aspect ratios. (1) In the **"Hook"** and **"Bun"** scenes, the Combined Reward guides the agent to retain more visual context (e.g., surrounding bananas) and avoid omitting object parts, which assists the downstream task model in scene understanding. (2) In the **"Flamingo"** scene, the Heuristic-only baseline produces an overly narrow box. Conversely, the Combined Reward encourages decomposing the region into three proper-sized boxes, avoiding distortion artifacts caused by resizing that would otherwise hinder the task model's recognition capability.

sensing model can effectively identify task-critical regions and enhance performance in both aerial and driving scenarios.

## H.4 SMALL OBJECT DETECTION ON LVIS

We further evaluate ACTIVE-O3 on the LVIS dataset and visualize its performance in Figure 16. Compared with alternative methods, ACTIVE-O3 demonstrates superior ability in selecting semantically meaningful regions for zoom-in, resulting in improved detection of small and rare object instances. These examples validate the general applicability of our approach to long-tail and fine-grained detection benchmarks.

## H.5 INTERACTIVE SEGMENTATION ON THINOBJECTS

We show in Figure 17 the performance of ACTIVE-O3 on the ThinObjects dataset for interactive segmentation. Our sensing model effectively identifies and focuses on regions with poor initial segmentation quality, enabling more precise refinement. These results highlight the utility of ACTI-

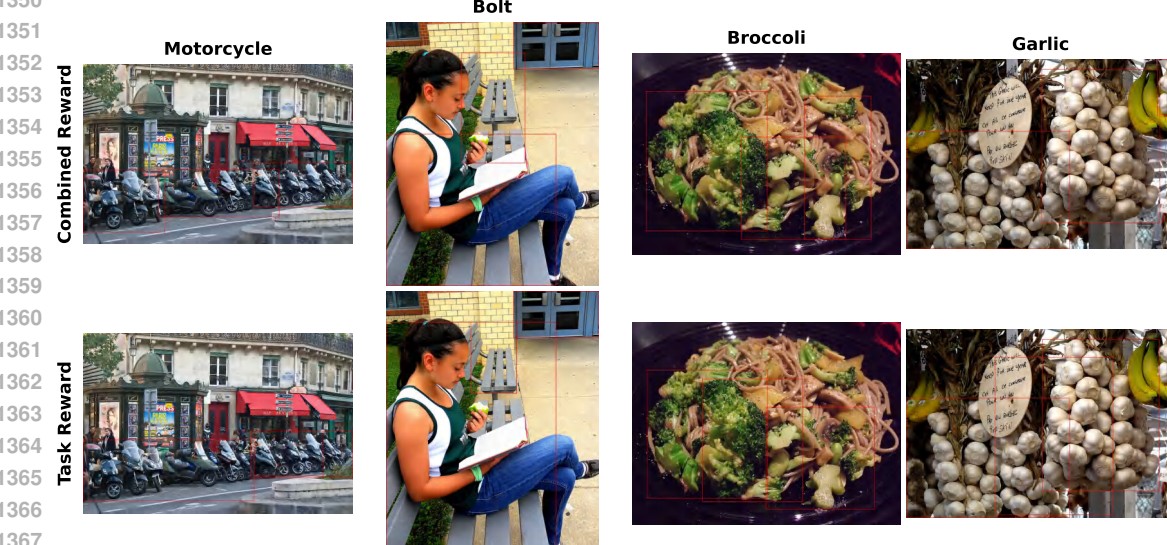

Figure 9: **Qualitative ablation of the Heuristic Reward component.** We compare our full method (Combined Reward, top row) with a baseline using only Task Reward (bottom row). (1) In the **"Motorcycle"** and **"Broccoli"** scenes, the Task-only baseline generates redundant boxes with significant overlap. Introducing the Heuristic Reward (Combined) effectively penalizes redundancy, leading to cleaner and more efficient proposals. (2) In the **"Garlic"** scene, this improved efficiency allows the agent to allocate resources to explore neglected areas, successfully detecting the previously missed garlic in the top-right corner.

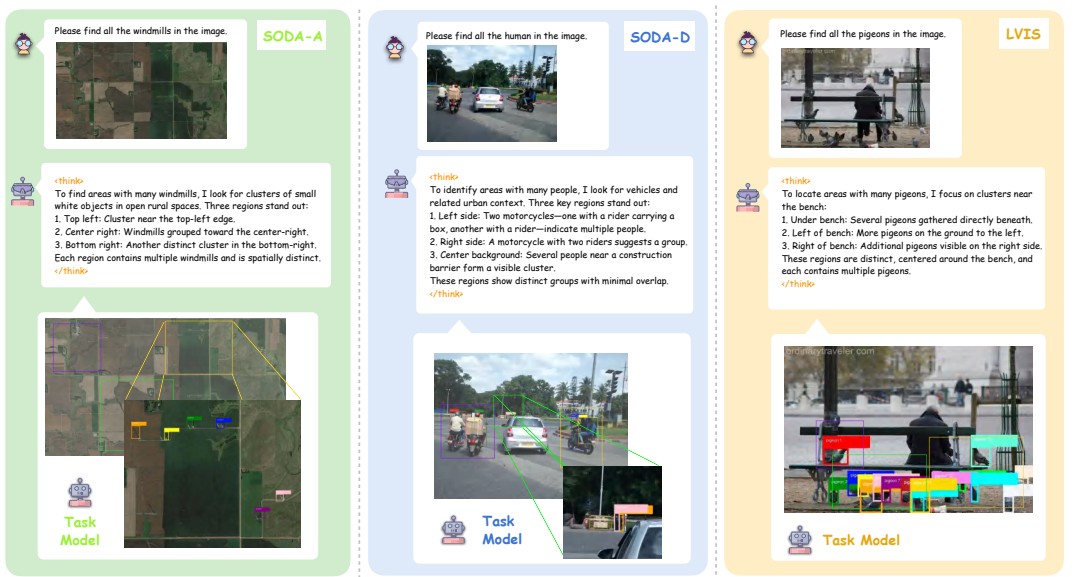

Figure 10: Visualization details of our proposed method on three datasets.

VE-O3 beyond detection, extending to segmentation tasks that require spatial reasoning and adaptive focus.

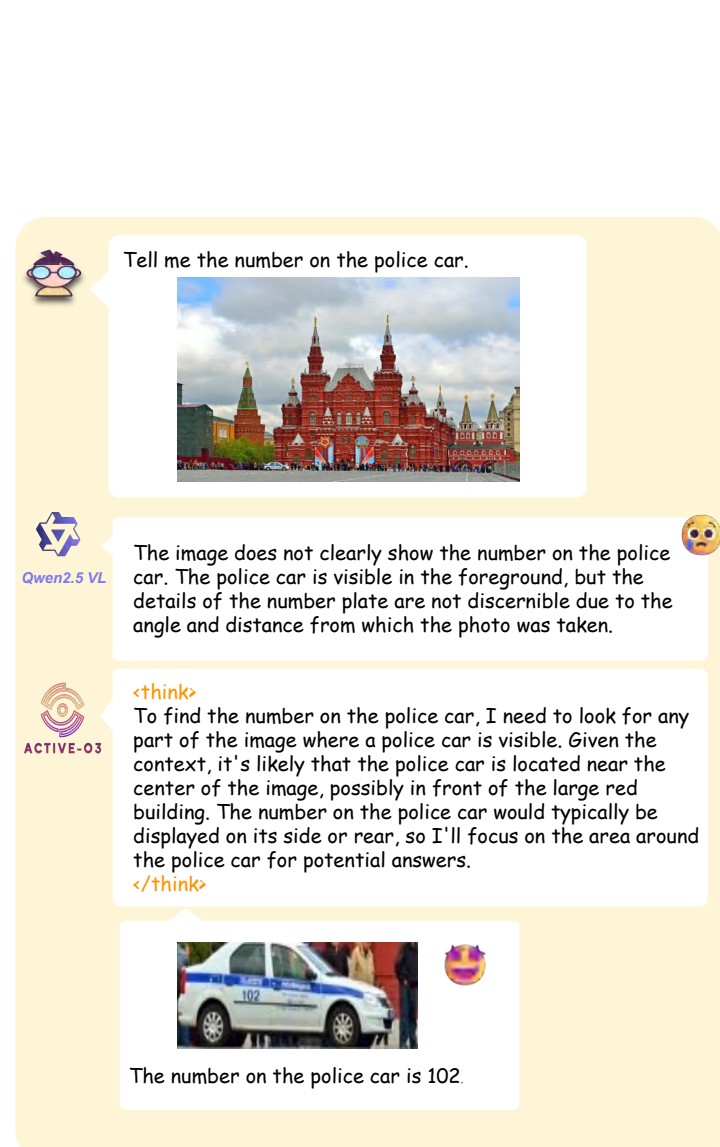

Figure 11: Zero-shot reasoning on the $V^*$ benchmark (Example 1). Given the question "Tell me the number on the police car", the baseline model (Qwen2.5 VL) fails to locate the relevant visual evidence due to limited resolution and reasoning capability. In contrast, our method (ACTIVE-O3) identifies the appropriate region through contextual reasoning and zoom-in selection. It successfully locates the number 102 on the police car, demonstrating strong spatial inference and fine-grained visual understanding.

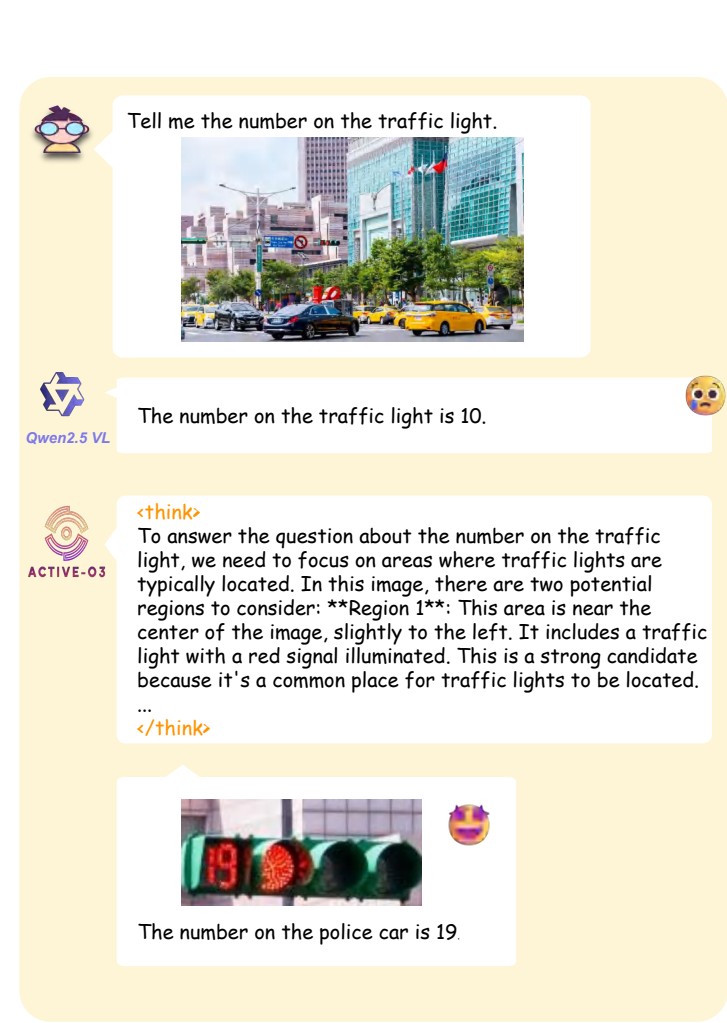

Figure 12: Zero-shot reasoning on the $V^*$ benchmark (Example 2). When asked "Tell me the number on the traffic light", Qwen2.5 VL incorrectly refers to unrelated text. In contrast, ACTI-VE-O3 locates and magnifies the precise area on the traffic light, accurately answering 10 through effective spatial localization.

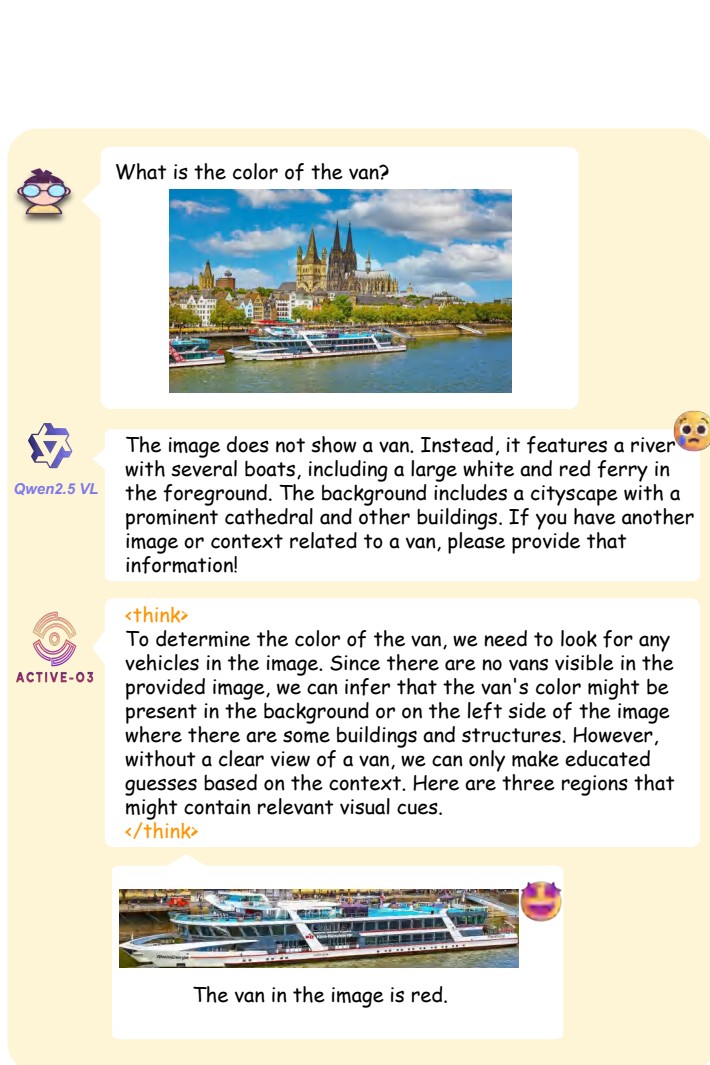

Figure 13: Zero-shot reasoning on the $V^*$ benchmark (Example 3). For the question "What is the color of the van?", the baseline model (Qwen2.5 VL) fails to detect the presence of the van and incorrectly claims that no such object is visible. In contrast, ACTIVE-O3 accurately identifies the small red van in the background and correctly answers red, demonstrating its ability to localize and reason over subtle visual cues that are easily overlooked.

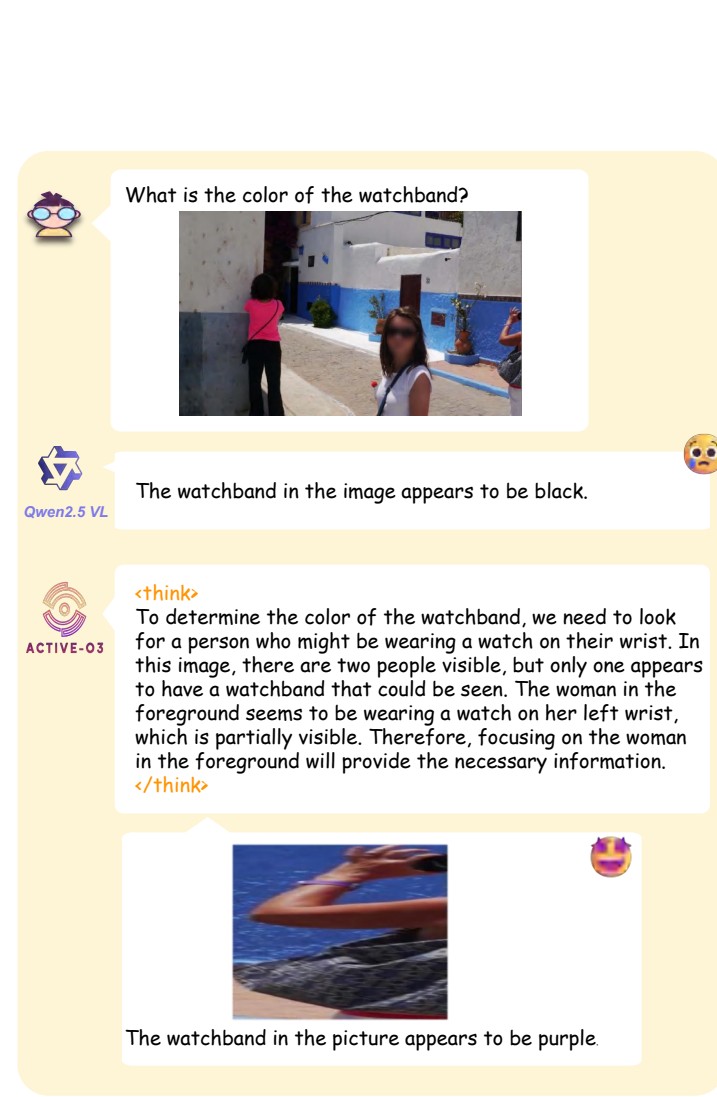

Figure 14: Zero-shot reasoning on the $V^*$ benchmark (Example 4). Given the question "What is the color of the watchband?", baseline predictions are inconsistent. ACTIVE-O3 focuses on the wrist of the foreground figure, providing the accurate answer (purple) by effectively zooming in on the fine-grained detail.

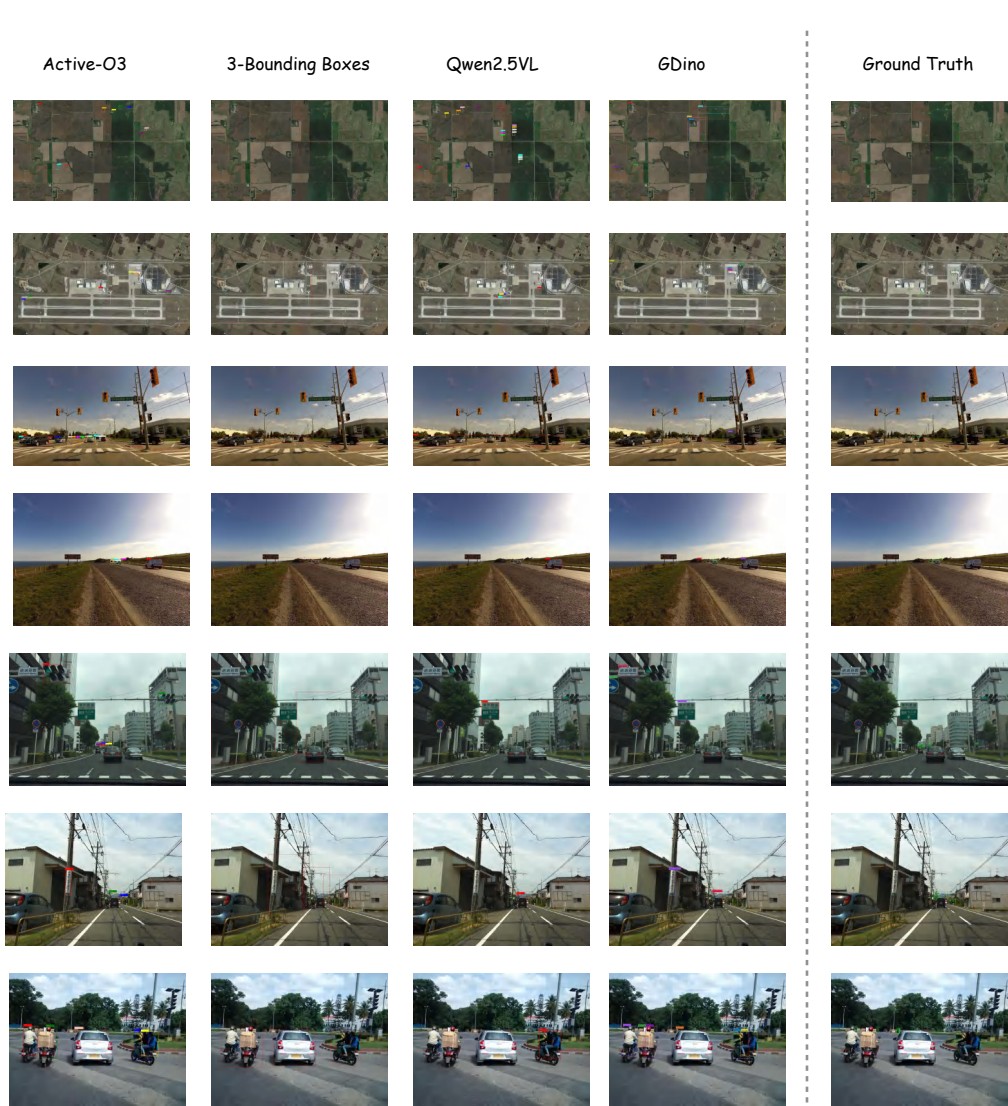

Figure 15: Visualization of Small Object Detection results on SODA-A and SODA-D datasets. Each row shows a different example from either SODA-A (top two rows) or SODA-D (remaining rows). The second column illustrates the candidate regions selected by our sensing model. Zoom in for better visibility of fine details and small objects.

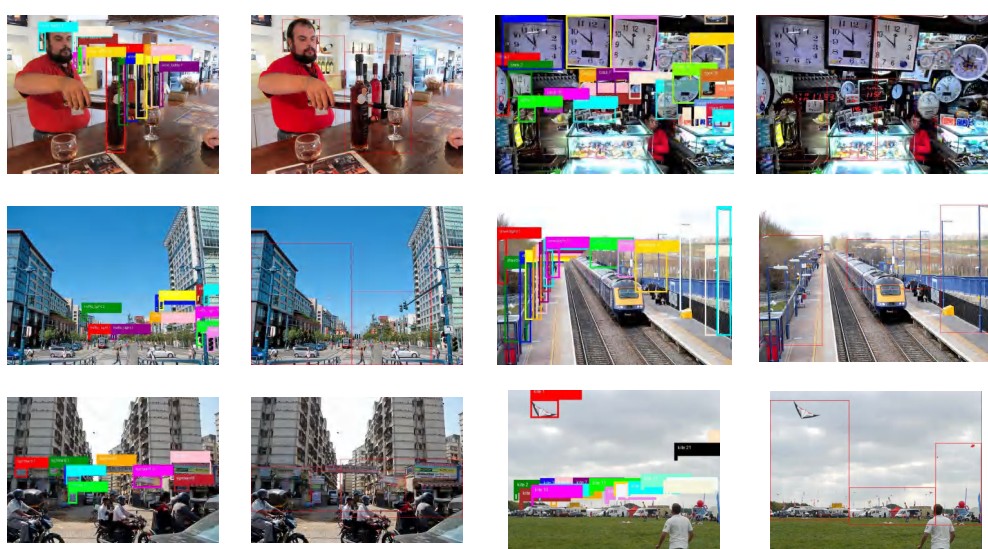

Figure 16: Visualization of object detection results on various scenes from the LVIS dataset. The left column shows the candidate regions selected by our sensing model.

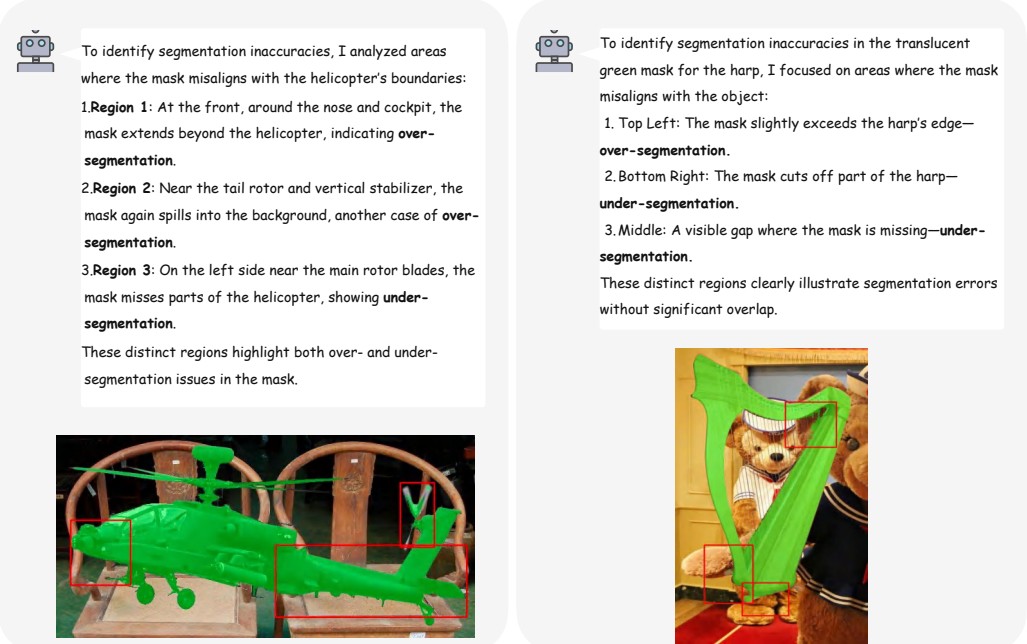

Figure 17: Interactive segmentation analysis on ThinObjects. ACTIVE-O3 identifies specific regions with segmentation inaccuracies by reasoning over visual cues. The left example (helicopter) reveals both over-segmentation (e.g., mask spilling beyond the nose and tail) and under-segmentation (e.g., missing rotor parts). The right example (harp) similarly highlights areas where the mask exceeds or misses the object boundary. These results demonstrate ACTIVE-O3's capability to localize fine-grained segmentation errors, facilitating efficient and targeted mask refinement.

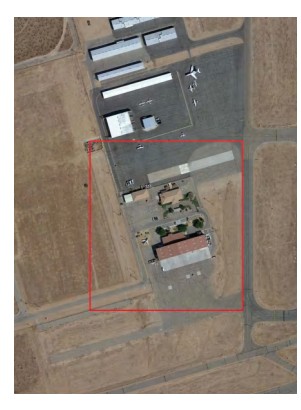
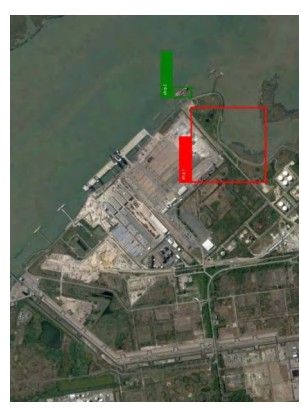

SODA-A

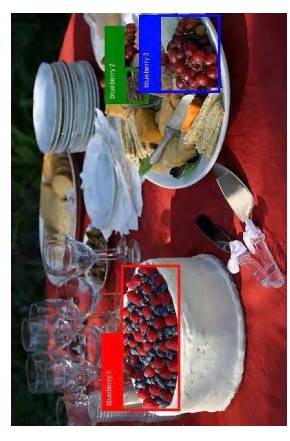
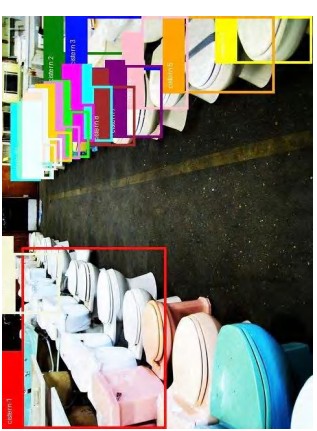
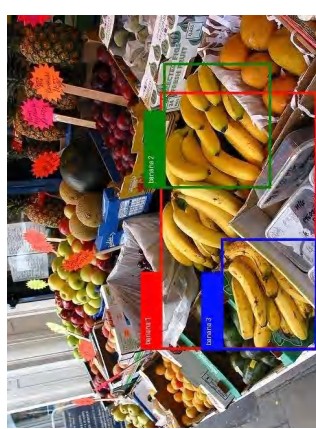

LVIS

