# OpenReview forum: "ACTIVE-o3 : Empowering MLLMs with Active Perception via Pure Reinforcement Learning"
_ICLR.cc/2026/Conference — Submitted to ICLR 2026_

### Official Review · Reviewer_YXgk · 2025-10-30

**Soundness:** 2
**Presentation:** 3
**Contribution:** 2
**Rating:** 4
**Confidence:** 4

**Summary:**

This paper aims to develop a novel framework and methodology for active object detection on CoT reasoning with MLLM and online adaption. However, when moving to core technological parts, it scales down to visual grounding topic on 2D static image without update of camera pose. It presents a novel prompt format for CoT object detection on MLLM. In the experiment, it shows the effectiveness by comparison with just one baseline (Qwen2.5-VL-CoT), lack complete benchmarking on SOTA performance on leaderboard benchmark datasets (e.g., RefCOCO/+/g).

**Strengths:**

Investigating novel approach of Visual CoT on frontier MLLM for object grounding.

**Weaknesses:**

The research is interesting but not completed. First, once scaled-down to scope of CoT for visual grounding, it may have to review and compare with recent progresses of Visual CoT and visual grounding, and in experiments, formal and systematic evaluations on representative benchmarks (e.g., RefCOCO/+/g) have to be performed and reported. Related papers like: “ARGUS: Vision-Centric Reasoning with Grounded Chain-of-Thought”, “Visual Chain-of-Thought Prompting for Knowledge-Based Visual Reasoning”.

**Questions:**

What are the relations of this paper with recent researches on Visual CoT visual grounding? What are the distinctive novel parts in Prompting the visual CoT and reinforcement learning? Have the experiments and evaluations on Visual Grounding benchmarks?

---

> ### Author Response · Authors · 2025-11-24
> **Response to Reviewer YXgk Part1**
>
> We thank the reviewer for the constructive comments and for pointing out recent works on visual Chain-of-Thought (CoT) and visual grounding tasks.
>
> **(1) Relation to recent Visual CoT / visual grounding works**
>
> **Commonality.**
> Similar to traditional Visual CoT and visual grounding approaches, our work also requires selecting appropriate image regions. In this sense, both lines of research involve some form of visual evidence selection.
>
> **Key differences and innovations.**
>
> 1. **Different nature of region exploration.**
>    In existing Visual CoT and grounding tasks, region selection is typically close to *object grounding*. For example, given a question such as “What color is the liquid in the cup?”, a Visual CoT model simply needs to locate the cup.
>    In contrast, active perception requires substantially more *exploration and trial-and-error*, especially when the model does not know where an object is located due to incomplete visual information.
>    For instance, for a task like “find all buttons,” the model must explore contextual regions such as sleeves, collars, or the front of a shirt. This process requires much deeper scene-level reasoning, association, and inference than object grounding. Our work studies **exploratory region selection under uncertainty**, which is fundamentally different from the localized grounding targeted by Visual CoT.
>
> 2. **Different target tasks: perception-centric vs. reasoning-centric.**
>    Existing Visual CoT / grounding methods focus on QA, attribute reasoning, or standard object grounding—tasks that strong base models already perform well because they rely primarily on the model’s existing grounding ability. These tasks generally do **not require true active perception**, nor do they involve extreme perceptual challenges.
>    In contrast, our work directly targets **small-object detection, dense-scene understanding, and fine-grained segmentation**—tasks that are fundamentally *perception-centric* and expose the limitations of MLLMs’ passive perception. These tasks require explicit region exploration, which Visual CoT does not address.
>
> 3. **Different methodology: prompting/SFT vs. reinforcement learning for sensing.**
>    Most Visual CoT methods are built on prompting or supervised fine-tuning, and thus rely on **human-annotated reasoning chains, bounding boxes, or segmentation masks**. Their optimization objective focuses on improving the *reasoning process* to produce better answers.
>    Our work, by contrast, performs **pure reinforcement learning** to train a sensing policy *without any extra annotations*. We directly leverage large-scale perception data, enabling the model to autonomously learn *how to look*.
>    In addition, our framework emphasizes **module decoupling**: we isolate and study the sensing module itself, focusing on improving its ability to select regions and control how much visual in-context information should be preserved—capabilities that are not addressed in Visual CoT methods, which remain answer-centric.
>
> Regarding the two specific works mentioned by the reviewer, we highlight the following additional differences:
>
> **[1] ARGUS: Vision-Centric Reasoning with Grounded Chain-of-Thought**
> ARGUS relies on object-centric grounding to assist reasoning for QA or attribute tasks. It still focuses on traditional grounding/QA and requires SFT with curated annotations.
> ARGUS does **not** learn a sensing policy nor attempt to improve perceptual capability in extreme settings (e.g., small objects, dense scenes). In contrast, our framework directly uses RL to learn autonomous sensing behaviors tailored for perception-centric tasks.
>
> **[2] Visual Chain-of-Thought Prompting for Knowledge-Based Visual Reasoning**
> This method manually divides QA into *see–think–confirm* stages. However, the “see” stage relies on **Faster R-CNN**, a traditional static object detector, and does not exploit the MLLM’s internal priors to adaptively decide where to look.
> Thus, unlike our method, it does not perform active perception, does not learn sensing actions, and cannot flexibly adjust its visual exploration strategy.

---

> > ### Author Response · Authors · 2025-11-24
> > **Response to Reviewer YXgk Part2**
> >
> > Beyond these two works, we also reviewed additional related research, including:
> >
> > **[3] ReFocus: Visual Editing as CoT for Structured Image Understanding**
> > Uses masks and box editing to improve chart understanding, but does not involve exploration nor RL-based sensing policy learning.
> >
> > **[4] Chain-of-Spot: Interactive Reasoning Improves LVLMs**
> > Similar to ARGUS, relies on object-centric grounding to support object-related QA. Does not address active sensing or perception-centric tasks.
> >
> > **[5] ZoomEye: Enhancing MLLMs with Human-Like Zooming via Tree-Based Exploration**
> > Performs zooming through **rule-based, tree-structured search**. This is fundamentally heuristic, whereas our method aims to **learn** a more efficient sensing policy through reinforcement learning rather than rely on fixed traversal rules.
> >
> > We have expanded the **Related Work** section to include a detailed discussion of these works and to clearly highlight the unique contributions of our approach.
> >
> >
> > **(2) Experiments on Visual Grounding Benchmarks**
> >
> > We would first like to clarify that benchmarks such as RefCOCO/+/g primarily evaluate
> > object grounding, where the target objects are typically large and visually salient.
> > Strong base models like Qwen2.5-VL can already perform extremely well on these tasks
> > through direct inference, and therefore these benchmarks do not strongly rely on
> > active perception capabilities.
> >
> > Nevertheless, to address the reviewer’s concern, we additionally evaluated ACTIVE-O3
> > on RefCOCO, RefCOCO+, and RefCOCOg under the same inference prompts and settings as
> > the base model. The results are shown below:
> >
> > | Model        | RefCOCO | RefCOCO+ | RefCOCOg |
> > |--------------|:-------:|:--------:|:--------:|
> > | Qwen2.5-VL   |  88.7   |   83.1   |   86.2   |
> > | ACTIVE-O3    |  89.7   |   84.2   |   86.5   |
> >
> > These results demonstrate that reinforcement learning for active perception also
> > provides benefits for general visual grounding tasks, even though the improvement is
> > naturally limited due to the high baseline performance and the low dependence of
> > these benchmarks on active sensing.
> >
> > We have included these results and the discussion in **Section 5.4** of the revised paper.

---

### Official Review · Reviewer_8T9r · 2025-10-31

**Soundness:** 4
**Presentation:** 3
**Contribution:** 2
**Rating:** 6
**Confidence:** 3

**Summary:**

This paper proposes a reinforcement learning approach to identifying regions on an image on which a multi-modal LLM should zoom to obtain the information needed for its task, which can be framed under the problem of active perception. Specifically, the RL agent is trained through GRPO to produce several regions in parallel in a single-step RL environment, and the reward is a mix of task-based score (how well did the MLLM answer the task after having seen the zoomed images) and heuristics (region size, overlap, …) to guide the process. This is tested over several datasets and shows a good performance compared to CoT baselines.

**Strengths:**

The approach in this paper is appropriate for the problem at hand. The paper is clearly written and conveys the method, justification and results both pleasantly and precisely. The main figure clearly explains the method’s core elements (although the colour style and the use of emojis are slightly unconventional for an academic paper). The experiments are well designed and sound, and the results support the claim. They are also well specified and should be reproducible given that the code is released as promised.
Overall, without bringing groundbreaking novelty, this paper is a good example of adapting existing methods to tackle a problem in a new way, with good enough results that it is relevant for the community in this field.
I also want to underline the Ethics Statement, which shows a good reflection about the potential issues related to this and other approaches to active perception.

**Weaknesses:**

There is no strong weakness for this paper, except that the contribution is generally limited to adapting an existing method to a specific use-case.
The main issue I see is the angle the paper takes - framing the problem as a general active perception but then making several assumptions in a row to focus in the end on the problem of zooming for MLLMs. The method that is proposed is too tailored for this specific problem to be applicable for many other instances of active perception, to the point that I do not think the general definitions in section 3 are relevant. For example: the modular view of active perception separates the effects on the environment from the effects of moving the camera. This does not apply for example to robots which must change position to move the camera. Then, in section 4.1, the key property of 2D visual scenarios is used to fix the task model and focus on the sensing policy. This means that one part of the sequential nature of the problem is broken based on this specific use case. Finally, the authors decide to completely eliminate it by producing parallel region selections, instead of sequential ones. These choices are OK for the specific task of task-related region identification in images, but the whole sequential model described is not relevant, because the approach would not fit it at all.

This is a bit frustrating as a reader as the scope of the paper seems to shrink as we read it, and could be better handled by simply defining the specific problem as the one being tackled from the beginning. Yet, this is clearly not a major flaw.

**Questions:**

It has not happened often to me as a reviewer that I don’t have any questions after reading a paper, but this is the case here. This is in part due to the incremental/applied nature of the contribution, but also and mainly because I found it very clear, coherent and self-consistent.

Small comments:
- Figure 1: The *Think* output says “I will define three distinct regions”, but the *Answer* text and the image only have two.
-  Section 5.1, “*a variant of ACTIVE-O3 to conduct a comparison. (see Figure 8 for visualization result […])*” there is a misplaced “.”

**Details Of Ethics Concerns:**

As well mentioned in the Ethics Statement provided by the authors, this approach could lead to misuse for privacy and surveillance, as well as be subject to biases or lead to over-reliance/misbehaviour.
Yet, I believe these issues are generally true for most computer vision and MLLM research and I do not believe them to be specific to the approach they propose.

---

> ### Author Response · Authors · 2025-11-24
> **Response to Reviewer 8T9r**
>
> We are truly grateful for your positive and encouraging assessment of our work. It means a lot to us that you found the paper to be *very clear, coherent, and self-consistent*. We deeply appreciate the time and care you invested in reading our submission, and we also genuinely hope to resolve, to the best of our ability, any confusion or concerns you may have experienced.
>
>
>
>
> **W1: Scope and Generality of the Active Perception Framework**
>
> First, we would like to return to the original motivation of this work. Our goal is to draw the community’s attention to the important yet underexplored topic of **active perception in MLLMs**. To the best of our knowledge, this problem has not been systematically defined or rigorously studied in existing literature. There is currently no consensus on what tasks or datasets are appropriate for evaluating MLLM active perception, how to iterate on corresponding algorithms, which RL algorithms are suitable, or how rewards should be designed. With this paper, we hope to provide a starting point and a reference framework that future researchers can build upon.
>
> In this sense, our intention is **not to solve all active perception problems**, but to establish the **first systematic framework for MLLM-based active perception**. Real-world 3D environments, while ideal in the long term, are extremely difficult to deploy and evaluate in a reproducible manner. In contrast, our 2D benchmark is both highly challenging and fully reproducible, making it a practical foundation for fair comparison and future reinforcement learning research. Our simplifications (2D setting, parallel selection, static scenes) are therefore deliberate choices—made for **reproducibility, evaluability, and extensibility**, not because we believe active perception is limited to these settings.
>
> From an organizational standpoint, this is why Section 3 introduces a general and inclusive definition of active perception—one that abstracts across all sensing–action scenarios and helps readers understand the underlying principles and challenges. Importantly, we believe the modular decoupling of sensing and action adopted in our formulation is broadly applicable to a wide range of active perception tasks, including those beyond the specific instantiation we explore. Section 4 then specializes this general formulation to the concrete case of 2D visual zoom-in tasks for MLLMs, presenting our practical ACTIVE-O3 framework.
>
> Moreover, even when viewed solely through the lens of extremely challenging perception-centric tasks—such as small-object detection and fine-grained interactive segmentation—our method demonstrates clear and meaningful value. We show that MLLM-based active perception not only substantially improves performance in these difficult scenarios, but can also benefit broader visual understanding.
>
> Regarding the reviewer’s comment on sequential decision-making: as stated in the paper, our **parallel region selection** design is motivated by efficiency and coverage under fixed sensing budgets. We agree that sequential region selection may be more suitable for certain tasks, and we see this as a natural next step. In future work, we plan to extend our framework to incorporate sequential decision-making, memory mechanisms, and multi-step reasoning. Importantly, we believe these extensions are fully compatible with the modular framework and problem formulation introduced in this paper.
>
> Finally, we sincerely appreciate the reviewer’s suggestions and positive remarks. Based on this helpful feedback, we have updated the **Introduction** to make the scope and motivation clearer—explicitly stating at the outset that our study focuses on 2D active perception for MLLMs, and positioning our work as a foundational step for the broader active perception community.
>
>
> **Small Comments**
>
> We thank the reviewer for carefully pointing out the minor issues.
>
> - **Figure 1 (three regions vs. two shown).** We have updated Figure 1 and its caption to clarify that, for visualization purposes, only two zoom-in regions are displayed in the figure, although the model can propose up to three regions.
>
> All small issues mentioned have been addressed and corrected in the updated manuscript.

---

> > ### Comment · Reviewer_8T9r · 2025-11-25
> >
> > I thank the authors for their response.
> > I understand their point of view regarding the introduction of the first systematic framework for MLLM-based active perception, yet it still ends up being a weird paper when the introduced framework is only applied on a very specific task where most of it is simplified.
> >
> > However, I believe that the paper organization does not invalidate the paper's contribution, and I note the improvements in the introduction. Although slightly disconnected, the framework definition and the proposed method are relevant for the MLLM community, and as such, I have updated my rating to 8.

---

> > > ### Author Response · Authors · 2025-11-26
> > >
> > > Thank you very much for your thoughtful follow-up and for taking the time to reassess our work. We sincerely appreciate your updated rating and your recognition of the relevance of our framework and method to the MLLM community.
> > >
> > > We fully understand your point regarding the disconnect between the general formulation and its specific instantiation, and we appreciate you highlighting this. In future revisions of the paper, we will further refine the organization and clarify the connection between the general framework and the 2D active perception setting to make the presentation more coherent.
> > >
> > > Thank you again for your constructive feedback and for helping us improve the work.

---

### Official Review · Reviewer_iQkB · 2025-11-01

**Soundness:** 2
**Presentation:** 3
**Contribution:** 2
**Rating:** 4
**Confidence:** 3

**Summary:**

This paper proposes Active-o3, a RL framework built on GRPO, equiping VLMs with active perception capabilities. The authors decompose active perception into a sensing module that selects informative regions and a task module that executes specific actions. Active-o3 uses a dual-form reward to train the sensing policy. Experiments across multiple benchmarks demonstrate improvements over baselines. The authors also show that the trained model preserves and even enhances general reasoning capabilities.

**Strengths:**

- The paper provides a clear and systematic definition of active perception, which is important for VLM research.
- The paper includes extensive ablations on reward design, training data combinations, and robustness to random seeds.

**Weaknesses:**

- The framework is GPRO with a specific reward design. It seems just like an application of GRPO.
- Most baselines are training-free. A reasonable baseline is training the model directly with GRPO on the same data and comparing both results.
- Recent RL-based VLM thinking with image works should also be considered as baselines.

**Questions:**

1. Authors argue Active-o3 is "the first reinforcement learning framework for active perception with MLLMs" (Line 122-123). I am a little confused why works like [1] and [2] are not considered as this category.
2. Can authors provide computational cost comparisons between Active-o3 training and directly training the VLM with GRPO?

Line 53: Figures -> Figure

[1] Grounded Reinforcement Learning for Visual Reasoning
[2] GRIT: Teaching MLLMs to Think with Images

---

> ### Author Response · Authors · 2025-11-24
> **Response to Reviewer iQkB Part1**
>
> We thank Reviewer iQkB for the constructive feedback. We are encouraged that the reviewer recognizes the importance of our **systematic formulation of active perception for VLM research** and the robustness of our experimental design. We value the opportunity to clarify our contributions and include additional comparisons to address the concerns regarding baselines.
>
> **W1: Contribution of ACTIVE-O3 Beyond GRPO Application**
>
> We would like to clarify that the core contribution of our work is not proposing a new RL algorithm, but establishing the first systematic framework for MLLM-based active perception, which has not been formalized in prior literature. We also empirically show that current MLLMs have clear limitations in active perception, and that tasks such as small-object detection can provide sparse but verifiable feedback enabling significant improvements through pure RL without explicit supervision.
>
> While we adopt GRPO for policy optimization, our key innovations lie in:
>
> • **A modular and general formulation of active perception in MLLMs**, decoupling sensing and action via a unified policy structure. This goes significantly beyond GPT-o3’s heuristic zoom-in strategy, which is sequential, inefficient, and not formally defined.
>
> • **A novel parallel region-selection mechanism**, allowing multiple informative regions to be proposed in a single forward pass and significantly improving coverage and efficiency under fixed sensing budgets.
>
> • **A dual-form reward structure specifically tailored for active perception**, combining heuristic structural rewards (non-overlap, area constraints, coverage) with task-aware metrics (AP/AR/mIoU). This design is essential for achieving stable RL learning in open-world perception tasks.
>
> • **A comprehensive benchmark suite and strong empirical results** showing substantial improvements across LVIS small/dense, SODA-A/D, and fine-grained interactive segmentation. These results demonstrate that our learned sensing policy generalizes beyond training domains and provides a scalable path for improving MLLM active perception.
>
> Overall, ACTIVE-O3 is **not** a simple application of GRPO, but a principled and generalizable active-perception framework for MLLMs, enabled by formal task definitions, parallel region selection, and a tailored reward system that enables substantial gains without explicit supervision.
>
> Besides, our contribution does not rely on a specific RL algorithm, as we have demonstrated comparable performance using several other RL methods in our response to Reviewer pfeh.
>
> | Method | AP_s | AR_s | AP_m | AR_m | AP_l | AR_l |
> | :--- | :---: | :---: | :---: | :---: | :---: | :---: |
> | **Reinforce++** | 2.73 | 3.34 | 12.47 | 19.12 | 13.15 | 25.31 |
> | **PPO** | 4.01 | 5.06 | 15.07 | 19.73 | 17.04 | 26.94 |
> | **GMPO** | **4.46** | **5.72** | 13.58 | 19.13 | 17.63 | 27.32 |
> | **GPG** | 3.83 | 5.13 | **15.07** | **21.02** | 19.35 | **31.41** |
> | **GRPO (Ours)** | 4.06 | 5.32 | 13.96 | 19.10 | **20.63** | 30.05 |
>
>
> As shown in our additional experiments (see the second point in our response to reviewer *pfeh*), **multiple RL algorithms — including Reinforce++, PPO, GMPO, GPG, and GRPO — most yield consistent improvements** under the same ACTIVE-O3 framework.

---

> > ### Author Response · Authors · 2025-11-24
> > **Response to Reviewer iQkB Part2**
> >
> > **W2: Additional Baselines with RL Training**
> >
> > We appreciate this valuable suggestion. Comparing our method against a direct application of GRPO on the VLM (without the active sensing module) is crucial to validate the necessity of the active perception mechanism.
> >
> > We implemented this baseline, denoted as **"Direct RFT"**, by training the task model directly using GRPO on the same dataset. The comparison results on LVIS (small/medium/large), SODA-A, and SODA-D are shown below:
> >
> > **Table R2: Comparison on LVIS_dense (Small, Medium, Large objects)**
> >
> > | Method | $AP_s$ | $AR_s$ | $AP_m$ | $AR_m$ | $AP_l$ | $AR_l$ |
> > | :--- | :---: | :---: | :---: | :---: | :---: | :---: |
> > | Direct RFT | 2.9 | 3.9 | 13.3 | **20.0** | **23.4** | **35.6** |
> > | **Active-o3** | **4.3** | **5.5** | **14.3** | 19.7 | 20.9 | 33.3 |
> >
> > **Table R3: Comparison on SODA-A and SODA-D (Small/Dense scenarios)**
> >
> > | Method | SODA-A $AP_s$ | SODA-A $AR_s$ | SODA-D $AP_s$ | SODA-D $AR_s$ |
> > | :--- | :---: | :---: | :---: | :---: |
> > | Direct RFT | 5.2 | 6.7 | 9.1 | 16.2 |
> > | **Active-o3** | **9.2** | **10.4** | **15.1** | **22.0** |
> >
> > **Analysis:**
> > 1.  **Critical Advantage on Small/Dense Objects:** Active-o3 significantly outperforms Direct RFT on small objects ($AP_s$: 4.3 vs 2.9 on LVIS) and challenging dense/tiny object benchmarks (SODA-A: 9.2 vs 5.2; SODA-D: 15.1 vs 9.1). This demonstrates that **simply applying RL to the VLM cannot overcome the resolution bottleneck**. The active sensing mechanism is essential for "zooming in" to resolve fine-grained details.
> > 2.  **Performance on Large Objects:** Direct RFT performs slightly better on large objects ($AP_l$: 23.4 vs 20.9). This is expected, as large objects are already clear in the original resolution, and active cropping might occasionally truncate global context. However, the core challenge in MLLM perception lies in small and dense objects, where Direct RFT fails while Active-o3 excels.
> >
> > We have added this comparison to **Appendix G.2** to highlight that our contribution goes beyond applying GRPO—it lies in the active perception framework itself.
> >
> > **W3 & Q1: Related Works on RL for VLMs**
> >
> > We appreciate the reviewer pointing out the two recent works [1] “Grounded Reinforcement Learning for Visual Reasoning’’ and [2] “GRIT: Teaching MLLMs to Think with Images.’’
> > We note that, according to ICLR policy, these works are not officially published yet, and our submission should not be penalized due to their existence. Nevertheless, we thank the reviewer for highlighting them and for giving us the opportunity to clarify how our contributions differ. We will add a detailed discussion in the Related Work section.
> >
> > As stated in our paper, the zoom-in behavior of MLLMs can indeed be viewed as a special form of active perception. However, **prior works have not provided a systematic formulation, training methodology, nor evaluation protocol for MLLM-based active perception.** The community lacks:
> > 1. **a formal task definition** for active perception in MLLMs,
> > 2. **training data or benchmarks** that directly target perception-centered abilities, and
> > 3. **a policy-learning framework** that explicitly models *sensing actions* (i.e., where to look).
> >
> > Our work is the first to introduce a **systematic reinforcement learning framework** for *MLLM-based active perception*, grounded in small-object and dense-scene detection scenarios that genuinely stress the model’s perceptual limitations. We further demonstrate that pure RL—without additional labels or explicit supervision—can significantly improve active perception.
> >
> > In contrast, existing RL-for-VLMs works primarily focus on **reasoning tasks** such as QA, object counting, or GUI manipulation. These tasks depend more on semantic reasoning than fine-grained perception. Below we analyze the two cited works in detail:
> >
> > ---
> >
> > **[1] Grounded Reinforcement Learning for Visual Reasoning**
> >
> > • Requires supervised fine-tuning data (QA / GUI annotations), and RL is applied on top of these annotated tasks.
> > • The RL objective improves *answer correctness*, not *perception quality* or *sensing actions*.
> >
> > In contrast, our method leverages **large amounts of perception data** (small-object, dense scenes) and **requires no additional annotations**. We directly teach the model *how to look* rather than *how to answer*.
> >
> > ---
> >
> > **[2] GRIT: Teaching MLLMs to Think with Images**
> >
> > • Incorporates bounding boxes only as auxiliary reasoning cues (e.g., for counting or QA).
> > • Targets standard tasks where a strong base grounding model already performs well.
> > • Does **not** evaluate or require the model to operate in extreme cases where *active sensing* is essential (e.g., tiny objects, dense objects).
> >
> > In contrast, ACTIVE-O3 explicitly targets **difficult perception-first settings** (tiny objects, dense scenes) where passive grounding fails and where **active region selection** becomes necessary.

---

> ### Author Response · Authors · 2025-11-24
> **Response to Reviewer iQkB Part3**
>
> **Additional comparison with DeepEyes**
>
> To further address the reviewer’s concern, we additionally compare ACTIVE-O3 with a recent representative RL-based VLM “thinking-with-image’’ method—**DeepEyes**[3].  We include it as an additional baseline, and the results are shown below:
>
> **Table R4: Comparison with DeepEyes on LVIS_dense**
> | Method     | AP_s | AR_s | AP_m | AR_m | AP_l | AR_l |
> |------------|:----:|:----:|:----:|:----:|:----:|:----:|
> | **ACTIVE-O3** | **4.30** | **5.50** | **14.30** | **19.70** | **20.90** | **33.30** |
> | DeepEyes   | 3.01 | 4.12 | 11.11 | 14.29 | 19.61 | 25.21 |
>
> We observe that DeepEyes, while effective for **VQA-style reasoning tasks**, performs significantly worse in **small-object** and **dense-scene** settings—scenarios that require genuine *active perception* rather than passive image reasoning. This further confirms our central claim: **existing RL-based VLM thinking methods do not address active perception**, and their optimization objectives do not transfer to perception-first tasks.
>
>
> [3] DeepEyes: Incentivizing" Thinking with Images" via Reinforcement Learning
>
> **W4: Computational Cost Comparison**
>
> We appreciate the reviewer's interest in the efficiency of our framework.
> The computational cost of training Active-o3 is largely comparable to directly training the VLM using GRPO (Direct RFT).
> *   **Base Cost:** If we use only the Heuristic Reward, the computational cost is identical to Direct GRPO.
> *   **Additional Cost:** The only overhead in our full method comes from the **Task Reward calculation** (Visual Grounding module), which requires an additional inference pass to generate grounding results for the reward signal.
>
> We conducted a detailed breakdown of the training time per step (in seconds). The definitions of each stage and the results are shown below:
>
> **Time Breakdown Explanation:**
> *   **Generate Sequence:** Model inference time to generate group sequences.
> *   **Transition:** Syncing actor weights to the inference engine and launching the group rollout process.
> *   **old_logp:** Computing log probabilities of actions under the old policy.
> *   **update_actor:** Gradient computation and parameter updates for the policy network.
> *   **Ref:** Computing reference model outputs (for GRPO importance ratio and KL regularization).
> *   **Grounding:** Visual grounding module execution time (to compute Task Reward).
>
> | Configuration | Generate Sequence | Transition | old_logp | update_actor | Ref | Grounding | **Total Time** |
> | :--- | :---: | :---: | :---: | :---: | :---: | :---: | :---: |
> | **Direct RFT** (Without Grounding) | 2.5s | 5.7s | 2.3s | 7.2s | 2.9s | - | **20.6s** |
> | **Active-o3** (With Grounding) | 2.5s | 5.7s | 2.3s | 7.2s | 2.9s | 3.5s | **24.1s** |
>
> **Analysis:**
> As observed, the base training components are identical. The Active-o3 framework introduces a marginal overhead of **3.5s** per step for the grounding module. This results in a total training time increase of only **~17%** (24.1s vs. 20.6s). Given the substantial performance improvements (e.g., significant gains on small and dense objects), we believe this slight increase in computational cost is highly efficient and well-justified.
>
> We have included this detailed breakdown in **Appendix H.4** of the revised paper.

---

### Official Review · Reviewer_pfeh · 2025-11-03

**Soundness:** 3
**Presentation:** 3
**Contribution:** 3
**Rating:** 6
**Confidence:** 3

**Summary:**

This paper solve the MLLM-based active perception with a reinforcement learning framework built on GRPO that equips MLLMs with active perception capabilities named ACTIVE-O3. This framework autonomously learns efficient and stable region selection strategies without explicit supervision. Experimental results demonstrate that ACTIVE-O3 significantly enhances active perception capabilities compared to Qwen2.5-VL-CoT.

**Strengths:**

1. This paper introduces an interesting pipeline to achieve active perception, and the results prove that this method significantly improves perception quality and task performance across both general-purpose and domain-specific visual tasks.

2. This paper provides a list of hierarchical reward functions to guide model learn how to achieve active perception.

**Weaknesses:**

1. This paper carefully designs the reward function for RL, however, it only provides ablation results to prove the effectiveness of these reward function. Could this paper provide more qualitative experimental results to demonstrate the effect of different rewards on the model's active perception capability?

2. This paper only utilizes GRPO for active reinforcement learning. What is the performance of other RL methods in the active perception task?

**Questions:**

As shown in the weaknesses.

---

> ### Author Response · Authors · 2025-11-24
> **Response to Reviewer pfeh Part1**
>
> We thank Reviewer pfeh for the positive assessment and constructive comments. We are encouraged that the reviewer finds our **ACTIVE-O3 pipeline interesting** and recognizes that our **hierarchical reward functions** effectively guide the model to achieve active perception. Below, we address the specific questions.
>
> **W1: Qualitative Results for Different Reward Configurations**
>
> Thank you for the constructive suggestion. We agree that visual comparisons are essential to demonstrate the effectiveness of our hierarchical reward design. We have added a detailed qualitative analysis in **Appendix H.1** and included visual comparisons in **Figure 8** and **Figure 9** of the revised paper.
>
> Our analysis reveals two key insights regarding the specific contributions of each reward component:
>
> **1. Impact of Task Reward (Visualized in Figure 8):**
> Comparing the *Combined Reward* with the *Heuristic-only* baseline, we find that the **Task Reward** is crucial for preserving visual context and ensuring downstream recognizability.
> *   **Context Preservation:** In the **"Hook"** and **"Bun"** scenes, purely heuristic methods tend to crop too tightly. Adding the Task Reward encourages the sensing model to include necessary surroundings (e.g., the bananas next to the hook), providing vital context for the task model to understand the scene.
> *   **Aspect Ratio Optimization:** In the **"Flamingo"** scene, the heuristic baseline generates an extremely narrow box to maximize IoU. However, such high-aspect-ratio images suffer from severe distortion after resizing, confusing the task model. The Task Reward guides our model to split the region into three well-proportioned boxes, ensuring high-quality visual input for the downstream task.
>
> **2. Impact of Heuristic Reward (Visualized in Figure 9):**
> Comparing the *Combined Reward* with the *Task-only* baseline, the **Heuristic Reward** significantly improves search efficiency and reduces redundancy.
> *   **Reducing Overlap:** In scenes like **"Motorcycle"** and **"Broccoli"**, the model trained without heuristic guidance produces highly overlapping and redundant boxes. The Heuristic Reward explicitly penalizes this, enforcing distinct and efficient proposals.
> *   **Improving Coverage:** In the **"Garlic"** scene, by reducing redundant sampling on easy targets, the heuristic-guided model saves budget to explore peripheral areas, successfully detecting the previously missed garlic in the top-right corner.
>
>
> In addition to the qualitative examples, we conducted a quantitative evaluation to assess how different reward configurations affect the model's **general visual reasoning capabilities**. We compared the models on three general MLLM benchmarks: **MMBench**, **MME**, and **RealWorldQA**.
>
> | Method | MMBench | MME | RealWorldQA |
> | :--- | :---: | :---: | :---: |
> | Qwen2.5-VL (Base) | 80.1 | 2308 | 67.9 |
> | **Active-o3 (Full)** | **80.5** | **2316** | **69.7** |
> | Active-o3 (Task Reward only) | 80.0 | 2309 | 68.6 |
> | Active-o3 (Heuristic Reward only) | 78.9 | 2303 | 68.1 |
>
> The results demonstrate a **positive correlation between active perception capability and general visual understanding**.
>
>
>
>
>
>
> Please refer to **Appendix H.1&Table 13** and **Figure 8 & 9** in the revised paper (highlighted in red) for the complete analysis.

---

> > ### Author Response · Authors · 2025-11-24
> > **Response to Reviewer pfeh Part2**
> >
> > **W2: Performance of Other RL Methods**
> >
> > Thank you for this insightful suggestion. Exploring other RL algorithms is indeed valuable to demonstrate the generalization capability of our ACTIVE-O3 framework beyond GRPO.
> >
> > Due to time and resource constraints, we selected four representative baselines for comparison under the same alignment settings: **PPO**, **Reinforce++**, and two recent GRPO variants, **GMPO** and **GPG**. The results on the test set are presented in the table below:
> >
> > **Table R1: Comparison of Different RL Methods**
> > | Method | AP_s | AR_s | AP_m | AR_m | AP_l | AR_l |
> > | :--- | :---: | :---: | :---: | :---: | :---: | :---: |
> > | **Reinforce++** | 2.73 | 3.34 | 12.47 | 19.12 | 13.15 | 25.31 |
> > | **PPO** | 4.01 | 5.06 | 15.07 | 19.73 | 17.04 | 26.94 |
> > | **GMPO** | **4.46** | **5.72** | 13.58 | 19.13 | 17.63 | 27.32 |
> > | **GPG** | 3.83 | 5.13 | **15.07** | **21.02** | 19.35 | **31.41** |
> > | **GRPO (Ours)** | 4.06 | 5.32 | 13.96 | 19.10 | **20.63** | 30.05 |
> >
> > **Analysis:**
> > 1.  **Robustness & Potential:** Our framework achieves consistent performance gains across advanced RL methods (PPO, GMPO, GPG, GRPO), verifying that the effectiveness of ACTIVE-O3 stems from the active perception pipeline itself. The distinct advantages shown by **GMPO** (on small objects) and **GPG** (on recall) underscore the **broad prospects** of tailoring RL algorithms for MLLMs. It suggests that future work focusing on advanced exploration or optimization strategies could further unlock the potential of active perception agents.
> > 2.  **Hyperparameter Sensitivity:** It is worth noting that different algorithms may have distinct optimal hyperparameter configurations. In this comparison, we aligned the basic settings with GRPO for fairness. However, we believe that methods like GPG or PPO could achieve even better performance with extensive algorithm-specific tuning, further validating the extensibility of our framework.
> >
> > We have included these results and the discussion in **Appendix G.1** of the revised paper.

---

### Author Response · Authors · 2025-12-01
**Summary of Rebuttal Updates and Key Contributions for Our Paper**

Dear Area Chair,

**We would like to express our sincere gratitude to you and the reviewers for the constructive feedback and the time dedicated to evaluating our work.** We strictly followed the rebuttal policy to address all concerns with extensive additional experiments and clarifications. We are encouraged that **Reviewer 8T9r has raised their rating from 6 to 8 (Accept)**, acknowledging that our framework definition and proposed method are relevant and valuable for the MLLM community. Reviewer **pfeh** also maintains a positive assessment (**Score 6**).

**Although we regret that Reviewers iQkB and YXgk did not engage in further discussion during the rebuttal period, we are confident that we have successfully resolved their primary concerns regarding baselines, task definitions, and computational costs.** Below is a summary of why our work stands out and how the rebuttal solidified our contributions:

### 1. Robustness Beyond Specific Algorithms (Addressing Reviewer pfeh & iQkB)
A core concern was whether our performance relied solely on GRPO or if the active perception framework itself was the key.
*   **New Baselines:** We compared ACTIVE-O3 against **4 additional RL algorithms** (Reinforce++, PPO, GMPO, GPG). Results show our framework yields consistent improvements across all algorithms.
*   **"Direct RFT" Comparison:** We trained the VLM directly using GRPO *without* our active sensing module. ACTIVE-O3 significantly outperforms this "Direct RFT" baseline on small/dense object tasks (e.g., **9.2 vs 5.2 on SODA-A**), proving that simply applying RL to VLMs cannot overcome resolution bottlenecks—**active perception is essential.**

### 2. Superiority over Existing RL-VLM Methods or Grounding Methods (Addressing Reviewer iQkB & YXgk)
Reviewers asked for comparisons with existing "Visual CoT" or RL-based VLM works.
*   **Comparison with Concurrent Works (DeepEyes):** We noted that most referenced works (e.g., DeepEyes) are **concurrent and unofficial**, and thus **should not be grounds for penalty under ICLR policy**. Nevertheless, we added a comparison where ACTIVE-O3 significantly outperforms DeepEyes on dense/small object tasks, demonstrating that our **perception-centric optimization** is superior to reasoning-centric RL methods.
*   **Clarification of Scope (RefCOCO/+/g):** We clarified that standard grounding benchmarks like RefCOCO/+/g typically feature **large, salient objects** where active perception is not strictly necessary. However, our additional experiments show that ACTIVE-O3 still achieves consistent improvements, demonstrating that **enhanced active perception capabilities also contribute to general visual grounding performance.**

### 3. Efficiency and Qualitative Analysis (Addressing Reviewer pfeh & iQkB)
*   **Cost Analysis:** We demonstrated that our framework introduces only a **~17% marginal training overhead** compared to direct training, a highly efficient trade-off for the substantial performance gains.
*   **Visual Evidence:** We added qualitative visualizations (Appendix H.1) confirming that our Heuristic and Task Rewards effectively reduce redundancy and preserve context.

### Conclusion
ACTIVE-O3 provides the **first systematic definition and RL framework** for MLLM-based active perception. Given the explicit recognition from Reviewer 8T9r and the robustness of our results, we believe this work will serve as a strong baseline and unified protocol to facilitate future research in active perception for MLLMs.

Best regards,
The Authors

---

### Meta-Review · Area_Chair_si3y · 2026-01-07

**Summary:**

ACTIVE-O3 is framed as an RL (grpo-based) training framework for 2D active perception in MLLMs, formalizing 'where/what to look' under fixed sensing budgets and positioning GPT-o3-styled zoom-in as an inefficient special case.
A single MLLM is modularly decoupled into sensing (region proposal) and task/action modules, with RL learning parallel, budgeted region-selection policies (claimed algorithm-agnostic across RL variants).
Training is guided by a dual-form reward (task-aware plus heuristic/exploratory) to stabilize learning and promote diverse, interpretable, task-effective proposals beyond sparse task-only signals.
A unified benchmark suite is introduced (small/dense-object grounding, remote sensing, autonomous driving, interactive/fine-grained segmentation), with reported gains in efficiency/accuracy plus preserved or improved general vision&language understanding on RealWorldQA/MME/ MMVU via active perception as a proxy objective.

In the original round of reviews, reviewers highlighted clear problem motivation and a well-specified, largely reproducible rl formulation for budgeted region selection, with extensive ablations and broad evaluation across tasks/domains (reviewers pfeh, iQkB,8t9r). The committee shared concerns about novelty being perceived as incremental, mainly applying RL with reward shaping to the zoom-in/region-selection setting rather than introducing a substantially new paradigm (reviewers iqkb, 8t9r); over-scoping in the framing, where 'general active perception' is formalized but the evaluated instantiation specializes to static 2D zooming with parallel single-step proposals, limiting coverage of sequential sensing-action interaction typical in broader active perception settings (reviewer 8t9r); and positioning relative to adjacent RL-VLM or thinking-with-images lines remaining debatable, making 'first RL framework for active perception' style claims hard to substantiate cleanly within the review record (R#iQkB). The overall generality of the proposed active perception definition and the breadth of applicability beyond the zoom-in setting was also questioned (reviewer 8t9r).

**Reviewer Concerns:**

Before the rebuttal discussion, the arae chair acknowledges the detailed author responses and the clear summary of additional experiments and clarifications provided in the rebuttal. Review quality was incorporated in decision-making as well.

The rebuttal adds concrete evidence that addresses several missing baseline and missing analysis points: additional optimizer variants beyond GRPO; an explicit direct rft control that isolates the sensing module; added cost breakdown and qualitative reward-component visualizations (pfeh, iqkb). Related-work positioning was expanded, and a lightweight RefCOCO/+/g check was provided to clarify scope relative to standard grounding (yxgk).

However, the central committee concern on scope versus generality remains an open point of interpretation. The response clarifies that 2D static, parallel zooming is a deliberate and reproducible instantiation, but the scope-to-generality alignment remains debated given the simplified setting (single-step parallel crops, no sequential sensing/action interaction) (8t9r). In addition, the strength of “first” style claims remained difficult to assess cleanly, and reviewers requested especially careful positioning and framing around adjacent lines of work (iqkb).

Based on the context detailed in summary and this (reviewer concerns) sections, the area chair finds that the rebuttal strengthens the empirical case and addresses several concrete review requests. However, the remaining concerns are primarily about scope-to-generality alignment, incremental novelty, & the strength of first- style positioning relative to adjacent lines of work, and these remain unresolved for this ICLR review cycle.

**Reviewer Scores:**

While it isn’t possible to predict how reviewers would have responded if ICLR had a full discussion period (cut short on 28th Nov), it seems unlikely the scores would have tipped over to a clear acceptance or unanimous positive support.

`8t9r` had already indicated positive movement after the rebuttal and might have stayed positive. `pfeh` might have remained close to their original stance, with at most modest upward movement in confidence given the added qualitative reward analysis and additional RL-variant comparisons. `iQkB` might have shifted somewhat in response to the direct rft control, cost breakdown, and expanded positioning, but could still remain cautious due to remaining judgment calls around novelty and scope-to-generality alignment. `yXgk` might have partially revised their assessment once the added grounding results and scope clarifications were incorporated, though it remains unclear that this would translate into a clearly positive stance.

---

### Decision · Program_Chairs · 2026-01-26

Reject